# AlchemistCoder: Harmonizing and Eliciting Code Capability by Hindsight Tuning on Multi-source Data

**Zifan Song**[1,2*]   **Yudong Wang**[2*]   **Wenwei Zhang**[2*]   **Kuikun Liu**[2]
**Chengqi Lyu**[2]   **Demin Song**[2]   **Qipeng Guo**[2]   **Hang Yan**[2]
**Dahua Lin**[2,3,4]   **Kai Chen**[2†]   **Cairong Zhao**[1†]
[1]Tongji University [2]Shanghai AI Laboratory
[3]MMLab, The Chinese University of Hong Kong
[4]HKGAI under InnoHK

https://internlm.github.io/AlchemistCoder

## Abstract

Open-source Large Language Models (LLMs) and their specialized variants, particularly Code LLMs, have recently delivered impressive performance. However, previous Code LLMs are typically fine-tuned on single-source data with limited quality and diversity, which may insufficiently elicit the potential of pre-trained LLMs. In this paper, we present *AlchemistCoder*, a series of Code LLMs with enhanced code generation and generalization capabilities fine-tuned on multi-source data. To achieve this, we pioneer to unveil inherent conflicts among the various styles and qualities in multi-source code corpora and introduce data-specific prompts with hindsight relabeling, termed *AlchemistPrompts*, to harmonize different data sources and instruction-response pairs. Additionally, we propose incorporating the data construction process into the fine-tuning data as code comprehension tasks, including instruction evolution, data filtering, and code review. Extensive experiments demonstrate that *AlchemistCoder* holds a clear lead among all models of the same size (6.7B/7B) and rivals or even surpasses larger models (15B/33B/70B), showcasing the efficacy of our method in refining instruction-following capabilities and advancing the boundaries of code intelligence. Source code and models are available at `https://github.com/InternLM/AlchemistCoder`.

## 1 Introduction

Closed-source Large Language Models (LLMs) like ChatGPT and GPT-4 [33, 34] exhibit impressive code intelligence by learning on large-scale and diverse code corpus, which also benefits many other applications, such as math reasoning [6], embodied control [25], and agent [46]. Since open-source LLMs [40] still lag behind closed-source LLMs [34] in this field, there has been growing interest in investigating the acquisition of code capabilities by developing specialized Code LLMs [35, 14].

The training of Code LLMs mainly goes through pre-training and fine-tuning stages [35]. Pioneer

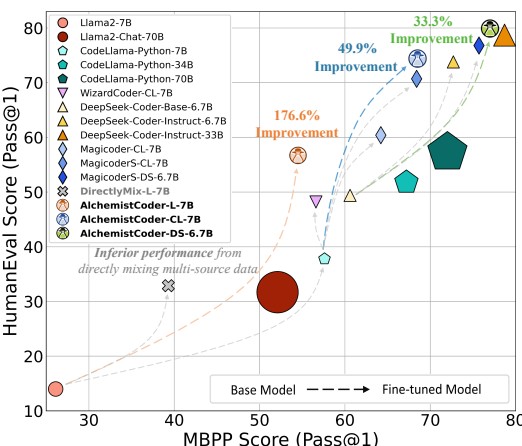

Figure 1: Performance scatter plot (*top right* is better) of open-source models on mainstream code benchmarks, HumanEval and MBPP. Our *AlchemistCoder* series demonstrates astonishing performance across all open-source Code LLMs.

---

*   Equal contributions
†   Corresponding author

38th Conference on Neural Information Processing Systems (NeurIPS 2024).

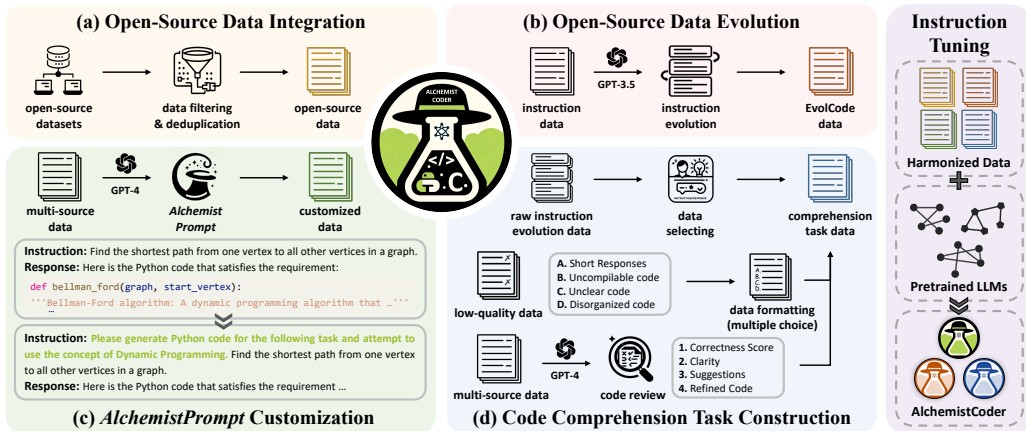

Figure 2: Overview for developing *AlchemistCoder* series. We first integrate high-quality open-source data (a) and conduct data evolution based on them (b). Then, we adopt *AlchemistPrompt* to harmonize their inherent conflicts (c) and construct code comprehension data (d). We use a mix of these data to fine-tune various pre-trained LLMs to obtain our *AlchemistCoder* models.

works [5, 32, 22, 36, 3] have amassed extensive code data for pre-training, while recent open-source models [30, 44] highlight the effectiveness of high-quality or targeted code fine-tuning datasets. Despite these advancements, current fine-tuning methods mainly rely on a specific type of code-related question-answering dataset, unlike the pre-training stage that integrates code-related corpus from various sources [35]. Such a discrepancy indicates that the fine-tuning data may lack the necessary diversity to fully stimulate the capabilities of base models, resulting in limited performance, generalization, and robustness.

To overcome the limitations in quality and diversity within single-source data, we pioneer to explore integrating multi-source data for Code LLM fine-tuning. However, this is a non-trivial paradigm and blindly integrating multi-source data can potentially lead to inferior performance (*e.g.*, the DirectlyMix-L-7B model in Fig. 1). To track this, we unveil inherent conflicts in multi-source code corpora, including conflicting code language requirements and response styles. Inspired by hindsight relabeling [1, 48], we propose to design data-specific prompts to harmonize the inherent conflicts for multi-source data integration, better eliciting the performance of base models. We term this form of prompts as *AlchemistPrompt*s, inspired by the power and definition of *Alchemists*:

*"Alchemist: Someone Who Transforms Things for the Better."* —— Merriam Webster

Specifically, we first integrate several open-source code datasets and conduct instruction evolution [30] based on some of them (Fig. 2(a, b)). As shown in Fig. 2(c), for instruction-response pairs of different sources, we adopt one LLM to generate *AlchemistPrompt*s that accurately and explicitly describe the characteristics as requirements of the response to enrich the instructions. In-depth, the efficacy of *AlchemistPrompt*s is twofold: 1) Harmonization between different data sources: *AlchemistPrompt*s generated from the same LLM have similar styles and can bridge the style differences between sources, while the introduction of *AlchemistPrompt*-customized data, accounting for only 5%, achieves a balance between data diversity and domain gaps; 2) Harmonization within instruction-response pairs: As fine-grained and data-specific prompts, *AlchemistPrompt*s are designed to augment instructions with specific programming languages, algorithm concepts, and other code-related information involved in responses, which can refine the alignment within instruction-response pairs and enhance the instruction-following abilities of fine-tuned models.

Apart from the conventional problem-solution data, we argue that the progression of code data (*e.g.*, data evolution, cleaning, and quality evaluation) reflects higher-level capabilities and offers valuable insights for the enhancement of Code LLMs. Consequently, we delineate the construction of data into three integral tasks for training: instruction evolution, data filtering, and code review (see Fig. 2 (d)), facilitating enhanced code comprehension capabilities.

We conduct extensive experiments with various base models [40, 35, 14] and develop the instruction-tuned *AlchemistCoder* series. As shown in Fig. 1, on two mainstream code benchmarks, HumanEval

**Instruction:**
Change the color of the text to blue.
## Input:
Text: The quick brown fox

**Response:**
To change the color of the text to blue, you can use CSS. Here's an example of how you can achieve this:

```html
<style>
    .blue-text {
        color: blue;
    }
</style>
<p class="blue-text">The quick brown fox</p>
```

By applying the "blue-text" class to the paragraph element, the text will be displayed in blue color.

*Inherent Conflict:*
*Missing Language Declaration.* The instruction does not specify the programming language, while the response assumes HTML/CSS and provides an implementation accordingly.

*AlchemistPrompt-customized Instruction:*
**Assume we are working with a HTML document and have the permissions to modify the CSS styles associated with it.**
Change the color of the text to blue.
## Input:
Text: The quick brown fox

---

**Instruction:**
Find the minimum of two given numbers.
## Input: 4, 6
## Output: The minimum number is 4

**Response:**
*Notebook Format:*
**Reasoning:**
To find the minimum of two given numbers, we can use an if-else statement. We will compare the two numbers, assign the smaller value to a variable called "minimum_number", and print out it.
**Code:**

```python
num1, num2 = 4, 6
if num1 < num2:
    minimum_number = num1
else:
    minimum_number = num2
print("The minimum number is", minimum_number)
```

**Execution Results:**
The minimum number is 4

*Inherent Conflict:*
*Format Exception.* The response introduces a notebook format not specified by the instruction.

*AlchemistPrompt-customized Instruction:*
**Complete the simple task in notebook format, including reasoning, code, and execution results.**
Find the minimum of two given numbers.
## Input: 4, 6
## Output: The minimum number is 4

---

**Instruction (Source #1&2):**
Replace all words with 5 letters or more that start with a vowel, and end with a consonant, with "*****" in the given text.
"The quick brown fox jumps over the lazy dog"

**Response (Source #1):**
Sure! Here is the text with the specified replacements:
"The quick brown fox jumps ***** the lazy dog"

**Response (Source #2):**
Here's a Python function that achieves this:

```python
import re
def replace_words(text):
    pattern = r'\b[aeiouAEIOU]\w{3,}[^aeiouAEIOU]\b'
    result = re.sub(pattern, '*****', text)
    return result

text = "The quick brown fox jumps over the lazy dog"
print(replace_words(text))
```

*Inherent Conflict:*
*Response Format Discrepancy.* For similar instructions, Source #1 provides the result directly, while Source #2 provides the implementation code, which may lead to confusion in understanding the instructions.

*AlchemistPrompt-customized Instruction (Source #2):*
**In this task, you are required to create a Python program that scans through a given text and replaces specified words.**
Replace all words with 5 letters or more that start with a vowel, and end with a consonant, with "*****" in the given text.
"The quick brown fox jumps over the lazy dog"

Figure 3: Examples of inherent conflicts (*e.g.*, various styles and quality) within multi-source code corpora. By applying *AlchemistPrompt*-customized instructions that are more consistent with the responses, the diversity from multiple sources can be effectively managed and utilized, thereby improving the quality of our fine-tuning data and the instruction-following capabilities of the fine-tuned models.

and MBPP, *AlchemistCoder* holds a clear lead among all models of equivalent size (6.7B/7B), and rivals or even surpasses larger models (15B/33B/70B), demonstrating harmonized and formidable code capabilities. Furthermore, we delve into the effectiveness of *AlchemistPrompt*s and discern that they alleviate the misalignment between instructions and responses within the data. Remarkably, *AlchemistPrompt*s allow the code corpus to also significantly improve the general capability of Code LLMs, as demonstrated by the improvements on MMLU, BBH, and GSM8K. Our main contributions are summarized as follows:

- Our work pioneers to integrate multi-source data for Code LLM fine-tuning to overcome the limitations of quality and diversity inherent in single-source data.

- We unveil inherent conflicts within multi-source code corpora and introduce *Alchemist-Prompt*s, revealing the power of hindsight tuning for code generation, aiming to harmonize the conflicts among sources and bridge the alignment within instruction-response pairs.

- We propose to incorporate data construction process into the fine-tuning data and design code comprehension tasks, including instruction evolution, data filtering, and code review, endowing comprehensive code capabilities.

- Extensive ablation and analytical studies confirm the efficacy of our key concepts for enhancing the diversity, quality, and cost-effectiveness of Code LLM fine-tuning data. Through instruction tuning on various base models, we develop the *AlchemistCoder* series, surpassing all Code LLMs of the same size on a wide spectrum of code benchmarks.

## 2 Method

To more comprehensively elicit the capability of the base LLMs, we first construct multi-source data for fine-tuning (§ 2.1), which is harmonized by *AlchemistPrompt*s to take effect(§ 2.2). Code comprehension tasks are also constructed to further improve the performance(§ 2.3). We also discuss the details and statistics of the filtered and harmonized multi-source data in § 2.4.

### 2.1 Multi-source data construction

To fully elicit the capability of code LLMs, we first collect the fine-tuning data from multiple sources (Fig. 2(a)) and adopt the instruction evolution [30] to improve the complexity of the instructions (Fig. 2(b)). However, integrating multi-source data for instruction tuning poses challenges. Naturally,

one code-related question can be solved by different coding languages with various algorithms or response styles (*e.g.*, with or without reasoning). When naively combing data curated by different developers with different LLMs, the model tends to learn to answer similar questions with different coding languages and response styles, as depicted in Fig. 3. On the one hand, learning diverse responses may elicit different capability aspects of the base models. On the other hand, since the learned responses to similar instructions often diverge due to implicit human intentions, the LLMs tend to be unaligned (to our expectation) after the fine-tuning on the directly mixed data (*e.g.*, we cannot expect which coding language the LLMs will use in real-world applications), resulting in inferior performance. Therefore, directly mixing multi-source data is not a promising solution and can be detrimental.

## 2.2 AlchemistPrompt

To harmonize the inherent conflicts within multi-source data, we propose to customize data-specific prompts called *AlchemistPrompt*s, (Fig. 2(c)), inspired by the concept of hindsight [1, 48]. Specifically, we employ GPT-4 [34] to play the role of an *Alchemist* and design the prompt as illustrated in Fig. 4 to obtain *AlchemistPrompt*s. For instance, for an instruction of 'Write code to find the shortest path from one vertex to all other vertices in a graph', if the response involves Python code of a Bellman-Ford algorithm with dynamic programming, we would expect to customize the instruction with an *AlchemistPrompt* of 'Please generate Python code for the following task and attempt to use the concept of Dynamic Programming'.

For the selection of data customized by *AlchemistPrompt*s, we calculate the differences in perplexities of generating responses with/without given instructions, called Conditional Perplexity Discrepancy (CPD). Then, we selectively chose data with higher CPD values for *AlchemistPrompt* harmonizations.

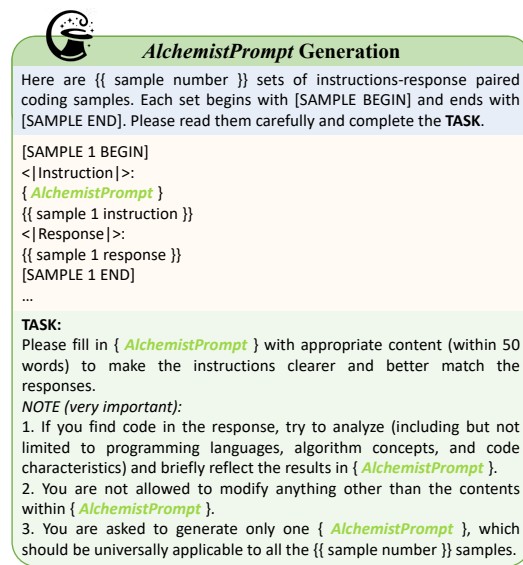

Figure 4: Detailed prompt designed for generating data-specific *AlchemistPrompt*s.

We treat CPD as an indicator of how data affects the complexity of model-generated responses under given conditions (*i.e.*, instructions), and its calculation formula is $\text{CPD} = \text{Perplexity}(\text{conditional\_instruction} + \text{response}) - \text{Perplexity}(\text{response})$. The level of CPD reflects the impact of the conditional instruction on the complexity of the generated response. Specifically, a high CPD indicates that the perplexity of the generated response significantly increases under the presence of a conditional instruction, which usually reflects a poor alignment between the instruction and the response, the instruction may be unclear or not specific enough, or insufficient contextual information, thereby increasing the difficulty of model response generation. By analyzing high CPD values, we can identify cases where instructions and responses are poorly aligned and more effectively optimize data quality.

The adjustments to data made by *AlchemistPrompt*s are relatively minor and well-calibrated. Our ablation study indicates that the optimal performance can be achieved by incorporating *AlchemistPrompt*s into only 5% samples, striking a balance between the diversity and domain gap resulting from the fusion of multi-source data. Crucially, by retrospectively analyzing responses and reinterpreting them as alternate goals, the *AlchemistPrompt*s serve to elevate the condition/goal of data. This hindsight integration [1, 48, 26] allows for a more nuanced and adaptive learning process, enhancing not only the models' comprehension of data but also refining instruction-following capabilities.

## 2.3 Code comprehension task

The existing training datasets for Code LLMs [24, 4, 39, 30, 44] primarily focus on the code generation task consisting of programming problems and their corresponding code solutions. However, we

contend that beyond this, the process of constructing code data demonstrates higher-level abilities. Consequently, we devise three code comprehension tasks relevant to data construction, including instruction evolution, data filtering, and code review (Fig. 2(d)).

**Instruction evolution.** Inspired by the concept of instruction evolution [45, 30], we employ GPT-3.5 [33] to construct instruction evolution task data, which entails augmenting the requirements for instructions and providing detailed explanations for programming tasks. Integrating the instruction evolution task aids the model in discerning the disparities before and after evolutions, thereby deepening the comprehension of programming requirements, code complexity, task decomposition, and other code-related concepts.

**Data filtering.** We identify four categories of low-quality data from multiple sources: (a) responses that are excessively short and lack code, (b) code that fails to compile, (c) code with poor clarity, and (d) code that does not adhere to the requirement in the instruction regarding its organization in function form. Each instruction in the data filtering task presents the model with a low-quality sample and prompts the model to classify it into one of the four categories. The data filtering task entails recycling the filtered-out data by offering counterexamples, thereby assisting the model in generating fewer low-quality responses.

**Code review.** In this task, we require the model to review a piece of code and assign scores between 0 and 10 for correctness and clarity separately. Additionally, the model is expected to provide suggestions for code improvement and present the refined code. To obtain higher-quality data, we utilize GPT-4 [34] to generate code reviews and select cases that are more representative, particularly those with average correctness and clarity scores exceeding 8 or falling below 6. Simultaneously, we focus on instances where one aspect exhibits severe deficiencies, *i.e.*, the score of correctness or clarity is equal to or below 4.

## 2.4 Data cleaning and decontamination

In practice, we have established a set of filtering rules to enhance our data cleaning and purification procedures. These rules involve excluding samples based on various criteria, such as response length (either too short or too long), absence of code or insufficient code content, non-compilable code, code failing test cases (pertinent to certain samples), responses structured in notebook form, and instances with excessive textual descriptions preceding the code. After conducting an extensive series of validation experiments, we conclusively decide to eliminate low-quality data meeting either of the following conditions: (a) responses that are excessively brief and lack code. Such responses typically offer direct answers to the instructions, neglecting both the code solution and explanatory annotations. Additionally, these samples frequently present overly simplistic questions in the instructions; (b) code solutions that are non-compilable or fail test cases (relevant to specific samples).

Concurrently, following [13], we employ N-gram similarity, cosine distance of code embeddings, and edit distance of code syntax trees to calculate the similarity between training data and samples in HumanEval and MBPP. We subsequently discard samples through this process of data filtering and deduplication, resulting in the removal of approximately 6% of the dataset.

## 2.5 Harmonized AlchemistCoder dataset

Our *AlchemistCoder* dataset (∼200M tokens) comprises four types of multi-source data, encompassing open-source datasets and three types of data constructed by us. Specifically, (a) open-source datasets including Evol-Instruct-Code-80k-v1 [10], CodeExercise-Python-27k [9], and evol-codealpaca-v1 [39], (b) EvolCode data generated from gpt-3.5-turbo following [30], (c) data customized by *AlchemistPrompt*s, and (d) data of the code comprehension tasks (*i.e.*, instruction evolution, data filtering, and code review).

We visualize the distributions of data sources and programming languages using two circular graphs in Fig. 5. Concurrently, Fig. 6 reports a distribution of text description lengths and code lines. Compared to CodeAlpaca [4] and OOS-INSTRUCT [44], our *AlchemistCoder* dataset presents a notably diverse distribution and maintains moderate overall text description and code lengths, benefiting significantly from the integration of multi-source data along with *AlchemistPrompt*s and code comprehension tasks. This is instrumental in contributing to a comprehensive evolution of code capability.

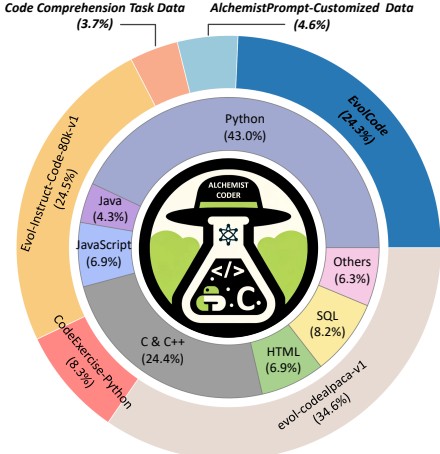

Figure 5: Data distribution analysis of our *AlchemistCoder* dataset. The outer and inner circular diagrams respectively display the distributions of data composition and programming languages. Data from *AlchemistPrompt*s and code comprehension tasks, constituting only 8% of the total data, plays a crucial role in harmonizing and polishing the fine-tuning data.

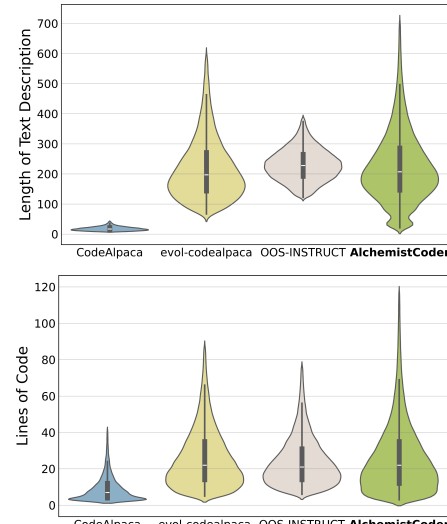

Figure 6: Comparative distribution of text description lengths (top) and code lines (bottom). Our dataset contains high-quality samples with more diverse distributions.

## 3 Experiments

In this section, we report results on various benchmarks of code generation and conduct ablation experiments. Furthermore, we present analytical studies to provide a more in-depth demonstration of the efficacy of our *AlchemistCoder*.

### 3.1 Benchmarks and implementation details

**Benchmarks.** We adopt six code benchmarks: HumanEval [5], HumanEval+ [27], HumanEval-X [49], MBPP [2], MBPP+ [27], and DS-1000 [21]. In addition, we access three mainstream benchmarks (MMLU [15], BBH [37], and GSM8K [8]) to evaluate generalization abilities. All evaluation and benchmark details can be found in Appendix §D.

**Baselines.** We compare with the following competitive baselines. Closed-Source Models: GPT-3.5-Turbo [33] and GPT-4-Turbo [34]. Open-Source Models: Llama 2 [40], CodeLlama [35], StarCoder [22], WizardCoder [30], DeepSeek-Coder [14], and Magicoder [44].

**Supervised fine-tuning.** We adopt Llama-2-7B, CodeLlama-Python-7B, and DeepSeek-Coder-Base-6.7B as the base models and fine-tune all the base models for 2 epochs using 32 NVIDIA A100-80GB GPUs. We set the initial learning rate, minimum learning rate, and optimizer warmup steps and at 1e-4, 6e-6, and 15, respectively. We use Adam optimizer [28] and choose a batch size of 2 with a sequence length of 8192.

### 3.2 Evaluation on code generation task

**Results on python code generation.** We first access HumanEval and MBPP to evaluate the capability of the *AlchemistCoder* series for Python code generation. These benchmarks necessitate models to generate code based on the function definitions and subsequently pass the test cases. Models are evaluated in zero-shot on HumanEval and 3-shot on MBPP. The comprehensive comparisons in Tab. 1 and Fig. 1 demonstrate the impressive capabilities of *AlchemistCoder* models. From the results, *AlchemistCoder-L* attains a remarkable performance boost of 42.7% and 28.4% pass@1 scores on HumanEval and MBPP respectively, compared to Llama 2-7B. Notably, *AlchemistCoder-DS* elevates the pass@1 scores to 79.9% and 77.0% on these benchmarks, holding an overall improvement of 33.3%. Moreover, our *AlchemistCoder* series with 7B parameters outperforms larger models (*e.g.*,

Table 1: Results of pass@1 on HumanEval (HumanEval+) and MBPP (MBPP+) benchmarks. We report the results of HumanEval and MBPP consistently from the EvalPlus [27] and the **bold** scores denote the best performance among models of the same size.

| Model | Params | Base Model | HumanEval (+) | MBPP (+) | Average (+) |
|---|---|---|---|---|---|
| *Closed-source Models* | | | | | |
| GPT-3.5-Turbo [33] | - | - | 72.6 (65.9) | 81.7 (69.4) | 77.2 (67.7) |
| GPT-4-Turbo [34] | - | - | **85.4 (81.7)** | **83.0 (70.7)** | **84.2 (76.2)** |
| *Open-source Models* | | | | | |
| Llama 2-Chat [40] | 70B | Llama 2 | 31.7 (26.2) | 52.1 (38.6) | 41.9 (32.4) |
| CodeLlama-Python [35] | 70B | Llama 2 | 57.9 (50.0) | 72.4 (52.4) | 65.2 (51.2) |
| CodeLlama-Instruct [35] | 70B | CodeLlama | **65.2 (58.5)** | **73.5 (55.1)** | **69.4 (56.8)** |
| CodeLlama-Python [35] | 34B | Llama 2 | 51.8 (43.9) | 67.2 (50.4) | 59.5 (47.2) |
| WizardCoder-CL [30] | 34B | CodeLlama-Python | 73.2 (56.7) | 73.2 (51.9) | 73.2 (54.3) |
| DeepSeek-Coder-Instruct [14] | 33B | DeepSeek-Coder-Base | **78.7 (67.7)** | **78.7 (59.7)** | **78.7 (63.7)** |
| StarCoder [22] | 15B | - | 34.1 (33.5) | 55.1 (43.4) | 44.6 (38.5) |
| CodeLlama-Python [35] | 13B | Llama 2 | 42.7 (36.6) | 61.2 (**45.6**) | 52.0 (41.1) |
| WizardCoder-SC [30] | 15B | StarCoder | **51.9 (45.7)** | **61.9** (44.9) | **56.9 (45.3)** |
| Llama 2 [40] | 7B | - | 14.0 (10.4) | 26.1 (17.5) | 20.1 (14.0) |
| StarCoder [22] | 7B | - | 24.4 (21.3) | 33.1 (29.2) | 28.8 (25.3) |
| CodeLlama-Python [35] | 7B | Llama 2 | 37.8 (33.5) | 57.6 (42.4) | 47.7 (38.0) |
| WizardCoder-CL [30] | 7B | CodeLlama-Python | 48.2 (42.1) | 56.6 (42.4) | 52.4 (42.3) |
| DeepSeek-Coder-Base [14] | 6.7B | - | 47.6 (41.5) | 70.2 (53.6) | 58.9 (47.6) |
| Magicoder-CL [44] | 7B | CodeLlama-Python | 60.4 (49.4) | 64.2 (46.1) | 62.3 (47.8) |
| MagicoderS-CL [44] | 7B | CodeLlama-Python | 70.7 (60.4) | 68.4 (49.1) | 69.6 (54.8) |
| Magicoder-DS [44] | 6.7B | DeepSeek-Coder-Base | 66.5 (55.5) | 75.4 (55.6) | 71.0 (55.6) |
| DeepSeek-Coder-Instruct [14] | 6.7B | DeepSeek-Coder-Base | 73.8 (69.5) | 72.7 (55.6) | 73.3 (62.6) |
| MagicoderS-DS [44] | 6.7B | DeepSeek-Coder-Base | 76.8 (65.2) | 75.7 (56.1) | 76.3 (60.7) |
| *AlchemistCoder-L (ours)* | 7B | Llama 2 | 56.7 (52.4) | 54.5 (49.6) | 55.6 (51.0) |
| *AlchemistCoder-CL (ours)* | 7B | CodeLlama-Python | 74.4 (68.3) | 68.5 (55.1) | 71.5 (61.7) |
| *AlchemistCoder-DS (ours)* | 6.7B | DeepSeek-Coder-Base | **79.9 (75.6)** | **77.0 (60.2)** | **78.5 (67.9)** |

Table 2: Results of pass@1 on HumanEval-X. We present the multilingual code capabilities of our *AlchemistCoder* with the respective base models and competitors (6.7B/7B).

| Model | Python | C++ | Go | Java | JS | Avg |
|---|---|---|---|---|---|---|
| Llama 2 | 14.0 | 11.0 | 6.1 | 11.0 | 14.0 | 11.2 |
| CodeLlama | 31.7 | 27.4 | 12.8 | 25.6 | 32.9 | 26.1 |
| *AlchemistCoder-L* | 56.7 | 31.1 | 25.6 | 45.1 | 41.5 | 37.1 |
| CodeLlama-Python | 37.8 | 33.5 | 30.5 | 39.6 | 35.4 | 35.4 |
| MagicoderS-CL | 68.3 | 47.6 | 39.6 | 34.8 | 57.9 | 49.6 |
| *AlchemistCoder-CL* | 74.4 | 53.1 | 42.7 | 64.0 | 52.4 | 57.3 |
| DeepSeek-Coder-Base | 47.6 | 45.1 | 38.4 | 56.1 | 43.9 | 46.2 |
| MagicoderS-DS | 72.6 | 63.4 | 51.8 | 70.7 | 66.5 | 65.0 |
| *AlchemistCoder-DS* | 79.9 | 62.2 | 59.8 | 72.0 | 68.9 | 68.6 |

Table 3: Pass@1 results of models with 6.7B/7B parameters on DS-1000. pd, np, tf, sp, skl, torch, and plt represent Pandas, Numpy, Tensorflow, Scipy, Sklearn, Pytorch, and Matplotlib, respectively.

| Model | pd | np | tf | sp | skl | torch | plt | All |
|---|---|---|---|---|---|---|---|---|
| Llama 2 | 2.4 | 7.3 | 6.7 | 6.6 | 2.6 | 1.5 | 7.7 | 5.0 |
| CodeLlama | 12.0 | **27.7** | 17.8 | **13.2** | 12.2 | **20.6** | 15.5 | 17.0 |
| *AlchemistCoder-L* | 13.4 | 22.7 | 31.1 | 11.3 | 25.2 | 8.8 | **29.0** | 20.2 |
| CodeLlama-Python | 16.2 | 16.4 | 15.6 | 17.9 | 20.0 | 22.1 | 38.7 | 21.0 |
| MagicoderS-CL | 25.1 | 40.9 | 35.6 | 29.3 | 36.5 | 38.2 | 51.0 | 36.7 |
| *AlchemistCoder-CL* | 30.9 | 43.6 | 46.7 | 30.2 | 37.4 | 41.2 | 52.3 | 40.3 |
| DeepSeek-Coder-Base | 21.3 | 35.0 | 26.7 | 23.6 | 34.8 | 25.0 | 34.8 | 28.7 |
| MagicoderS-DS | 30.6 | 46.8 | 44.2 | 30.2 | 33.0 | 29.7 | 45.2 | 37.1 |
| *AlchemistCoder-DS* | 32.0 | 51.7 | 44.5 | 33.1 | 38.4 | 33.8 | 49.8 | 40.5 |

WizardCoder-CL-34B and CodeLlama-Instruct-70B) and rivals with GPT-3.5-Turbo, significantly bridging the performance gap between closed-source and open-source models.

**Results on multilingual code generation.** We compare the pass@1 accuracy of the base models and the corresponding fine-tuned *AlchemistCoder* models on Humaneval-X [49]. The results presented in Tab. 2 demonstrate that the *AlchemistCoder* series exhibits great improvements (exceeding 50%) for multilingual code generation, delivering comprehensive code capabilities.

**Results on code generation for data science.** We further conduct the evaluation of data science code completion on DS-1000 [21]. According to Tab. 3, *AlchemistCoder* models exhibit a notable improvement of up to 19.2% in overall performance compared to the base models. Particularly,

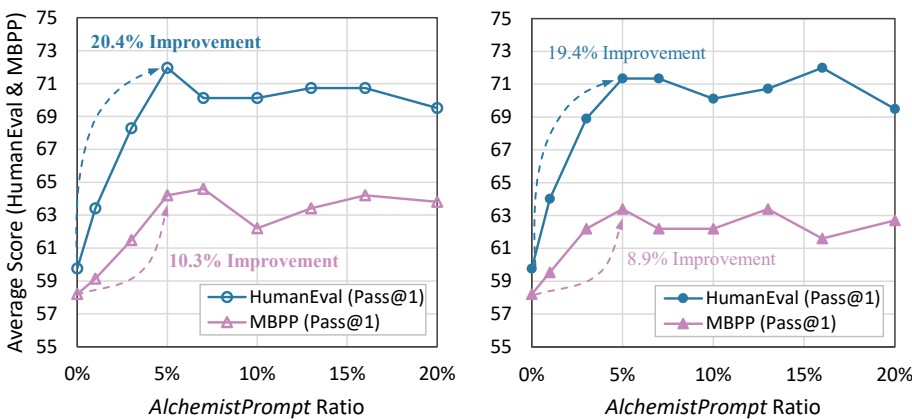

Figure 7: Ablation study on the proportion of *AlchemistPrompt*-customized data conducted on *AlchemistCoder-CL-7B*. Left: Augment the original data. Right: Replace the original data.

Table 4: Ablation study on the effectiveness of multi-source harmonization (*i.e.*, Multi-source Integration, Data Decontamination, and *AlchemistPrompt* Harmonization) and code understanding tasks (*i.e.*, Instruction Evolution Task, Data Filtering Task, and Code Review Task) for the *AlchemistCoder-CL-7B* model, evaluated on the HumanEval and MBPP benchmarks.

| Multi-source Integration | Data Decontamination | *AlchemistPrompt* Harmonization | Instruction Evolution Task | Data Filtering Task | Code Review Task | HumanEval (Pass@1) | MBPP (Pass@1) |
|---|---|---|---|---|---|---|---|
| - | - | - | - | - | - | 37.8 | 57.6 |
| ✓ | - | - | - | - | - | 54.6 (16.8↑) | 57.9 (0.3↑) |
| ✓ | ✓ | - | - | - | - | 59.8 (5.2↑) | 58.2 (0.3↑) |
| ✓ | ✓ | ✓ | - | - | - | 72.0 (12.2↑) | 63.4 (5.2↑) |
| ✓ | ✓ | ✓ | ✓ | - | - | 71.3 (0.7↓) | 65.8 (2.4↑) |
| ✓ | ✓ | ✓ | ✓ | ✓ | - | 73.8 (2.5↑) | 67.7 (1.9↑) |
| ✓ | ✓ | ✓ | ✓ | ✓ | ✓ | **74.4** (0.6↑) | **68.5** (0.8↑) |

*AlchemistCoder-CL* achieves an astonishing overall accuracy of 40.3% with relatively better performance in all libraries, demonstrating powerful capabilities in data science workflows.

## 3.3 Ablation study

**The Recipe of *AlchemistPrompt*s.** As illustrated in Sec. 2.2, *AlchemistPrompt*s can further align the instructions and responses of data samples and harmonize the domain gap between multiple sources. Code data from different sources may vary significantly in language style and content, including question types, code style, presence of comments, test cases, etc. Therefore, multi-source data mixing is a double-edged sword: it provides necessary diversity but can also bring large domain gaps. Adding concise corpus generated from the same Alchemist model (*i.e.*, *AlchemistPrompt*s with similar language styles) to a small amout of data can effectively bridge this gap while maintaining diversity. To find the appropriate recipe of *AlchemistPrompt*s that maintains a balance between data diversity and domain gap, we conduct ablation experiments on the proportion (0% to 20%) of data customized by *AlchemistPrompt*s. We adopt two settings: (a) augment the original data with its customized variant and report the results of fine-tuning for 2 epochs on CodeLlama-Python-7B; (b) replace the original data and report the results of fine-tuning for the same steps (*i.e.*, keeping the number of tokens used consistent). As shown in Fig. 7, *AlchemistCoder* is particularly enhanced when the proportion of customized data increases from 1% to 5%, and nearly peaks in performance at 5%. Thus, we introduce *AlchemistPrompt*s into 5% of the training set to balance the performance gain and the generation cost. Additionally, both two strategies effectively enhance the performance and validate the efficacy of our approach. To push the limit of *AlchemistCoder*, we employ the augmentation strategy in our performance experiments. In addition, we present detailed experimental results from the multi-source integration and harmonization process in our Appendix §C to offer a more in-depth demonstration of the *AlchemistPrompt*s efficacy as data complexity scales.

**Efficacy of the code comprehension tasks.** We conduct an ablation study on the key components of the code comprehension tasks to ascertain their individual contributions to the overall performance. As reported in Tab. 4, compared to the baselines (the first and second rows), the model demonstrates

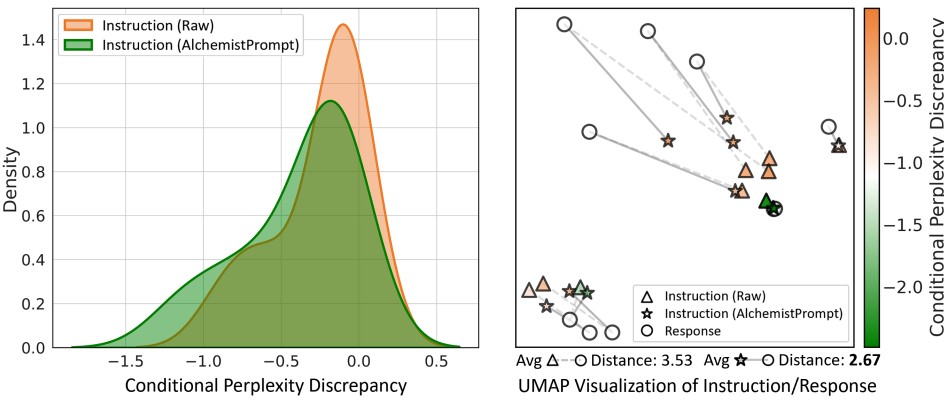

Figure 8: In-depth analysis of the efficacy from *AlchemistPrompt*s. Left: Kernel Density Estimation of Conditional Perplexity Discrepancy. Right: UMAP visualization of 10 instruction/response groups.

enhanced performance on both benchmarks following the incremental incorporation of code comprehension task data. Notably, the improvement (5.1% regard to the pass@1 metric) is particularly remarkable on MBPP. This indicates the significant contribution of all code comprehension tasks to furthering programming capabilities.

### 3.4 Analytical study

***AlchemistPrompt*s harmonize the discrepancy between instructions and responses.** To in-depth verify the efficacy of *AlchemistPrompt*s, we calculate the Conditional Perplexity Discrepancy (CPD, refer to Sec. 2.2) values of our fine-tuning data harmonized by *AlchemistPrompt*s, *i.e.*, the difference between $\text{Perplexity}(\text{conditional\_instruction} + \text{response})$ and $\text{Perplexity}(\text{response})$. The CPD value quantifies the difficulty change in generating responses before and after adding specific inputs (*e.g.*, instructions) to the model (the smaller the value, the easier it becomes). Specifically, we adopt the instructions before and after customization by *AlchemistPrompt*s for comparison, and provide the Kernel Density Estimation of CPD in Fig. 8. Clearly, the latter (green) gains smaller overall CPD values, indicating that *AlchemistPrompt*s are beneficial for prediction and can provide effective contextual information. Furthermore, we randomly select 10 groups of these samples and use UMAP [31] to map their feature representations into a 2-D space in the right of Fig. 8. From the fact that the solid lines are generally shorter than the dashed lines, our *AlchemistPrompt*s can harmonize the discrepancy between instructions and responses, leading to higher-quality data for attaining improved instruction-following ability.

***AlchemistCoder* models are better generalists.** To further analyze the comprehensive capabilities of our *AlchemistCoder*, we conduct evaluations on more diversified benchmarks, including MMLU [15] for multitask language understanding, BBH [37] for comprehensive reasoning, and GSM8K [8] for mathematical ability. The results are presented in Tab. 5 and illustrate that the *AlchemistCoder* models exhibit an overall performance increase of 6.4%, 13.6%, and 14.5% over the base models Llama 2, CodeLlama-Python, and DeepSeek-Coder-Base, respectively. Notably, CodeLlama-Python presents inferior performance on these benchmarks relative to Llama 2, indicating the discrepancy between natural language processing and code capabilities of open-source models. Such divergence can be attributed to "catastrophic forgetting" [11, 29, 20], often occurring when fine-tuning is exclusively concentrated on domain-specific data. By leveraging harmonized multi-source data, our *AlchemistCoder* series models achieve enhanced reasoning abilities, better instruction-following abilities, and improved context understanding, which contribute to develop better generalists.

**Error case analysis.** To meticulously dissect the improvements brought by our method, we provide an analysis of error cases on HumanEval and MBPP. We compare models before and after the introduction of *AlchemistPrompt*s and code understanding task data. The bar chart shown in Fig. 9 (top) indicates that these two types of key data help to better handle compilation errors (*i.e.*, SyntaxError, NameError, and ValueError), and eliminate the occurrence of no code written in the responses. On the other hand, the results of Fig. 9 (bottom) on MBPP suggest that the *AlchemistCoder* series incorporated with these two types of data attains stronger programming logic, as evidenced by the clear reduction in the 'Wrong Answer' error cases.

Table 5: Results of models (6.7B/7B) on various benchmarks, including MMLU for multitask language understanding, BBH for comprehensive reasoning, and GSM8K for mathematical ability.

| Model | MMLU | BBH | GSM8K | Avg |
|---|---|---|---|---|
| Llama 2 | 41.1 | 34.6 | 16.8 | 30.8 |
| CodeLlama | 31.5 | **42.7** | 14.4 | 29.5 |
| *AlchemistCoder-L* | 43.9 | **42.7** | 25.0 | 37.2 |
| CodeLlama-Python | 26.1 | 26.7 | 6.6 | 19.8 |
| MagicoderS-CL | 33.0 | **41.5** | 18.8 | 31.1 |
| *AlchemistCoder-CL* | 42.1 | 39.3 | 20.2 | 33.9 |
| DeepSeek-Coder-Base | 34.0 | 12.8 | 22.0 | 22.9 |
| MagicoderS-DS | 34.4 | 43.8 | 14.3 | 30.8 |
| *AlchemistCoder-DS* | 38.5 | 45.6 | 28.0 | 37.4 |

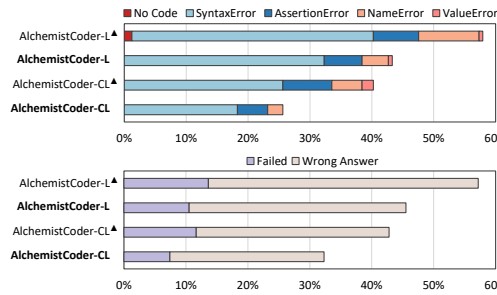

Figure 9: Analysis of error case proportions on HumanEval (top) and MBPP (bottom). ▲ represents the models fine-tuned without *Alchemist-Prompt*s and the code comprehension task data.

## 4   Related Work

**Code large language models.** Early researches [5, 32, 22] focus on collecting massive amounts of code data to develop pretrained Code LLMs. Recent efforts [30, 47, 44] are dedicated to fine-tuning these pretrained models with specific instructional data to further the coding abilities. For instance, WizardCoder [30] and Magicoder [44] construct their instruction tuning datasets based on CodeAlpaca [4] and the stack [18] dataset, respectively. In this work, we develop the *AlchemistCoder* series by instruction tuning on optimized multi-source data instead of single-category data as in previous methods, endowing astonishing and harmonized code capability.

**Instruction tuning.** Instruction tuning aims to enhance LLMs via fine-tuning pre-trained LLMs using samples of instruction/response pairs. Obtaining high-quality data for instruction tuning is typically challenging and extensive works have been dedicated to this endeavor. For instance, Alpaca [38] employs self-instruct [42] to generate instruction-following demonstrations. WizardLM [45] introduces Evol-Instruct and transforms the instruction data into more complex variants. In addition to Evol-Instruct, we also incorporate the data construction process itself as a form of data into the training. Moreover, although previous works [16, 43, 41, 23] utilize multiple fine-tuning datasets, we harmonize multi-source data at a fine-grained level.

**Learning from hindsight.** The concept of learning from hindsight [26] has been explored in goal-conditioned learning [17, 12]. Hindsight Experience Replay (HER) [1] is designed to re-label rewards and facilitate learning from sparse feedback. Korbak *et al.* [19] study the influence of human preferences during pre-training, showing improved performance when models are aligned with human preferences. Previous work primarily serves as an alternative to RLFT, utilizing HER to leverage (suboptimal) historical data for model learning. We focus on harmonizing the inherent conflicts within multi-source data through hindsight, to fully tap into the potential of base models.

## 5   Conclusion

In this paper, we propose an effective framework for integrating multi-source data to fine-tune Code LLMs, addressing the limitations in quality and diversity inherent within a single-source dataset. This is a non-trivial paradigm and we pioneer to unveil inherent conflicts in multi-source code corpora. To resolve this challenge, we innovatively design data-specific *AlchemistPrompt*s, inspired by hindsight relabeling. Additionally, we make the first effort of integrating the data construction process as code comprehension tasks into the training process. These key concepts enhance the diversity, quality, and cost-effectiveness of code fine-tuning data, facilitating the development of the *AlchemistCoder* series models with significantly improved and comprehensive coding capabilities.

## 6   Acknowledgments

This work is supported by National Natural Science Fund of China (62076184, 62473286) in part by Shanghai Natural Science Foundation (22ZR1466700). Besides, this project is funded in part by the Hong Kong Generative AI Research and Development Center (HKGAI) under the Innovation and Technology Commission (ITC)'s InnoHK. Dahua Lin is a PI of HKGAI under the InnoHK.

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

# Appendix

In the Appendix sections, we discuss limitations (§A), ethical considerations and broader impacts (§B), additional experimental results (§C), benchmark and evaluation details (§D), and more details of our *AlchemistCoder* fine-tuning data (§E).

## A    Limitations

Currently, GPT-4 holds an advantage in generating high-quality responses, and thus has been chosen as our *Alchemist* model. Compared to methods that heavily rely on GPT-4 to generate entire new datasets, we have been striving to minimize our dependence on GPT-4. Instead of using GPT-4 to generate data from scratch, we optimize a small amount of data. In the ablation experiments shown in Figure 7, we have verified that achieving optimal performance only requires using GPT-4 to generate *AlchemistPrompt*s for 5% of the data. Furthermore, the generation tasks we designed only require very short responses (less than 50 words), significantly reducing the token usage of GPT-4. Despite these efforts, the generation of *AlchemistPrompt*s is still a significant cost. We will explore fine-tuning open-source models to achieve the free generation of *AlchemistPrompt*s in the future. For data bias, Our *AlchemistPrompt*s enhance the instruction-following abilities of models, which can potentially mitigate biases. For example, if the model has a bias towards responding with Python code when the programming language is not specified, the inclusion of programming language declarations in *AlchemistPrompt*s helps to alleviate this bias. We have not delved into the aspect of data bias yet and will explore it in the future.

## B    Ethical Considerations and Broader Impacts

We use publicly available datasets, benchmarks, and models for training and evaluation, free from any possible harm toward individuals or groups. The generated data are relevant to code-related tasks and no personal identification information is involved. Furthermore, we adopt ChatGPT to polish the writing and assist with language. For broader impacts, *AlchemistCoder* enhances code generation and generalization through multi-source fine-tuning, promising improved software development efficiency, democratization of programming, and educational benefits. However, it also raises concerns about malicious use, intellectual property issues, and skill degradation. To ensure the responsible release of *AlchemistCoder* models, we will implement controlled access, provided usage guidelines, and engaged with the research community, thereby mitigating the risks of misuse or dual-use.

## C    Additional Experimental Resluts

### C.1    Details of fine-tuning tokens

In Tab. A1, we provide details of the training corpus used for fine-tuned Code LLMs.

### C.2    Data complexity and multi-source integration

Our research demonstrates that integrating data from multiple sources significantly increases data complexity and diversity, as evidenced by the broader distributions of code and description lengths shown in Figure 6 of the manuscript. While this integration facilitates the model's ability to learn richer feature representations, it also heightens the demands on the model to manage inputs of varying styles, formats, and quality. *AlchemistCoder* addresses this challenge by introducing *AlchemistPrompt*s, which help conduct harmonizations across various data sources and within instruction-response pairs. To offer a more in-depth demonstration of the *AlchemistPrompt*s efficacy as data complexity scales, we present detailed experimental results from the multi-source integration and harmonization process in Tab. A2

## D    Benchmark and Evaluation Details

### D.1    HumanEval/HumanEval+

Table A1: Pass@1 results on HumanEval (HumanEval+) and MBPP (MBPP+) benchmarks. *The column of FT Tokens (Source) specifies the tokens and origin of fine-tuning (FT) data used to train each model.*

| Model | Params | Base Model | FT Tokens (Source) | HumanEval (+) | MBPP (+) | Average (+) |
|---|---|---|---|---|---|---|
| ***Closed-source Models*** | | | | | | |
| GPT-3.5-Turbo | - | - | - | 72.6 (65.9) | 81.7 (69.4) | 77.2 (67.7) |
| GPT-4-Turbo | - | - | - | **85.4 (81.7)** | **83.0 (70.7)** | **84.2 (76.2)** |
| ***Open-source Models*** | | | | | | |
| Llama 2-Chat | 70B | Llama 2 | - | 31.7 (26.2) | 52.1 (38.6) | 41.9 (32.4) |
| CodeLlama-Python | 70B | Llama 2 | - | 57.9 (50.0) | 72.4 (52.4) | 65.2 (51.2) |
| CodeLlama-Instruct | 70B | CodeLlama | 5B (Llama Generation) | **65.2 (58.5)** | **73.5 (55.1)** | **69.4 (56.8)** |
| CodeLlama-Python | 34B | Llama 2 | - | 51.8 (43.9) | 67.2 (50.4) | 59.5 (47.2) |
| WizardCoder-CL | 34B | CodeLlama-Python | 80M (GPT Generation) | 73.2 (56.7) | 73.2 (51.9) | 73.2 (54.3) |
| DeepSeek-Coder-Instruct | 33B | DeepSeek-Coder-Base | 2B (GitHub Crawling) | **78.7 (67.7)** | **78.7 (59.7)** | **78.7 (63.7)** |
| StarCoder | 15B | - | - | 34.1 (33.5) | 55.1 (43.4) | 44.6 (38.5) |
| CodeLlama-Python | 13B | Llama 2 | - | 42.7 (36.6) | 61.2 (**45.6**) | 52.0 (41.1) |
| WizardCoder-SC | 15B | StarCoder | 80M (GPT Generation) | **51.9 (45.7)** | **61.9** (44.9) | **56.9 (45.3)** |
| Llama 2 | 7B | - | - | 14.0 (10.4) | 26.1 (17.5) | 20.1 (14.0) |
| StarCoder | 7B | - | - | 24.4 (21.3) | 33.1 (29.2) | 28.8 (25.3) |
| CodeLlama-Python | 7B | Llama 2 | - | 37.8 (33.5) | 57.6 (42.4) | 47.7 (38.0) |
| WizardCoder-CL | 7B | CodeLlama-Python | 80M (GPT Generation) | 48.2 (42.1) | 56.6 (42.4) | 52.4 (42.3) |
| DeepSeek-Coder-Base | 6.7B | - | - | 47.6 (41.5) | 70.2 (53.6) | 58.9 (47.6) |
| Magicoder-CL | 7B | CodeLlama-Python | 90M (GPT Generation) | 60.4 (49.4) | 64.2 (46.1) | 62.3 (47.8) |
| MagicoderS-CL | 7B | CodeLlama-Python | 240M (GPT Generation) | 70.7 (60.4) | 68.4 (49.1) | 69.6 (54.8) |
| Magicoder-DS | 6.7B | DeepSeek-Coder-Base | 90M (GPT Generation) | 66.5 (55.5) | 75.4 (55.6) | 71.0 (55.6) |
| DeepSeek-Coder-Instruct | 6.7B | DeepSeek-Coder-Base | 2B (GitHub Crawling) | 73.8 (69.5) | 72.7 (55.6) | 73.3 (62.6) |
| MagicoderS-DS | 6.7B | DeepSeek-Coder-Base | 240M (GPT Generation) | 76.8 (65.2) | 75.7 (56.1) | 76.3 (60.7) |
| ***AlchemistCoder-L (ours)*** | 7B | Llama 2 | 200M (GPT Harmonization) | 56.7 (52.4) | 54.5 (49.6) | 55.6 (51.0) |
| ***AlchemistCoder-CL (ours)*** | 7B | CodeLlama-Python | 200M (GPT Harmonization) | 74.4 (68.3) | 68.5 (55.1) | 71.5 (61.7) |
| ***AlchemistCoder-DS (ours)*** | 6.7B | DeepSeek-Coder-Base | 200M (GPT Harmonization) | **79.9 (75.6)** | **77.0 (60.2)** | **78.5 (67.9)** |

Table A2: Ablation study on the efficacy of multi-source integration and *AlchemistPrompt* harmonizations, evaluated on the HumanEval (Pass@1) and MBPP (Pass@1) benchmarks.

| Method | HumanEval | MBPP |
|---|---|---|
| Baseline (Llama2-7B) | 14.0 | 26.1 |
| *1*-source data fine-tuning | 18.3 | 29.0 |
| + *AlchemistPrompt* **harmonizations** | 22.6 (4.3↑) | 30.2 (1.2↑) |
| *2*-source data fine-tuning | 35.4 | 30.6 |
| + *AlchemistPrompt* **harmonizations** | 39.0 (3.6↑) | 32.8 (2.2↑) |
| *3*-source data fine-tuning | 37.8 | 35.4 |
| + *AlchemistPrompt* **harmonizations** | 43.9 (6.1↑) | 40.8 (5.4↑) |
| *4*-source data fine-tuning | 40.2 | 42.2 |
| + *AlchemistPrompt* **harmonizations** | **55.1 (14.9↑)** | **49.4 (7.2↑)** |

HumanEval [5] and HumanEval+ [27] are benchmarks for assessing LLMs' code generation, focusing on functional correctness. HumanEval+ expands on HumanEval by significantly increasing test cases through EvalPlus, using LLM and mutation strategies for more rigorous evaluation. This approach reveals performance drops in models like GPT-4 and ChatGPT against challenging tests, emphasizing the need for diverse test scenarios to accurately evaluate LLMs' coding abilities. For evaluation on HumanEval and HumanEval+, we adopt the prompt designed for HumanEval/HumanEval+ tasks shown in Fig. A1. Fol-

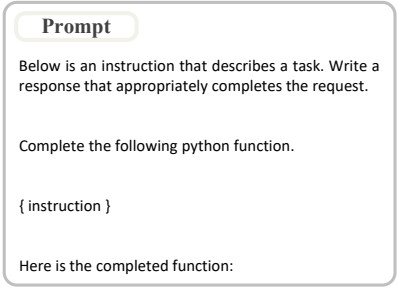

Figure A1: Prompt used to evaluate on HumanEval and HumanEval+.

lowing prior works [49, 7, 44], we use the greedy decoding
strategy and focus on comparing the pass@1 metric.

## D.2 MBPP/MBPP+

The MBPP (Mostly Basic Python Programming) benchmark [2] consists of around 1,000 Python
challenges, crowd-sourced to test basic programming skills, including fundamentals and standard
library use. Aimed at beginners, each challenge offers a description, solution, and three tests for
verifying solution accuracy. MBPP+ [27] is an extension of the MBPP benchmark, utilizing a subset
of hand-verified problems from MBPP-sanitized to ensure tasks are well-defined and unambiguous.
For the evaluation on MBPP and MBPP+, we adopt the three-shot prompt shown in Fig. A2.

## D.3 HumanEval-X

HumanEval-X [49] is a comprehensive benchmark that assesses the capabilities of code generation
models across multiple programming languages, including Python, C++, Java, JavaScript, and Go.
It consists of 820 meticulously created data samples, each accompanied by test cases, making it
an invaluable resource for evaluating and improving multilingual code generation models. The
benchmark aims to provide insights into the models' proficiency in solving diverse coding challenges
and their accuracy in generating functionally correct code in different languages. For evaluation on
HumanEval-X, we do not use specific prompts and follow the original test prompts.

## D.4 DS-1000

The DS-1000 benchmark [21] adapts 1000 different data science coding problems each with unit
tests from StackOverflow and checks both execution semantics and surface-form constraints. These
realistic problems are drawn from seven popular data science libraries in Python, including Matplotlib
(plt), NumPy (np), Pandas (pd), SciPy (scp), Scikit-Learn (sk), PyTorch (py), and TensorFlow (tf).
DS-1000 has two modes: completion and insertion, and here we only evaluate completion, as the
basic CodeLlama-Python does not support insertion. For evaluation on DS-1000, we do not use
specific prompts and follow the original test prompts.

## D.5 MMLU

The Massive Multitask Language Understanding (MMLU) benchmark [15] is an evaluation frame-
work designed to measure the depth and breadth of knowledge that LLMs possess. It accomplishes
this by testing these models across 57 varied tasks in both zero-shot and few-shot scenarios. The tasks
encompass a wide array of topics, including basic math, American history, computer science, law,
and more, challenging the models to leverage their acquired knowledge to solve complex problems.
MMLU seeks to emulate the multifaceted way in which human knowledge and problem-solving
skills are assessed, offering a comprehensive gauge of a model's ability to understand and apply
information across multiple domains. For evaluation on MMLU, we do not use specific prompts and
follow the original test prompts.

## D.6 BBH

The BIG-Bench Hard (BBH) Benchmark [37] is a specialized evaluation framework tailored to
rigorously test the capabilities of LLMs. This benchmark targets a selection of tasks that have
historically proven challenging for LLMs, focusing on areas where models typically do not exceed
average human performance. The BBH Benchmark aims to push the boundaries of what LLMs can
achieve by emphasizing complex reasoning, deep understanding, and nuanced interpretation, setting
a high bar for model development and performance evaluation. For evaluation on BBH, we do not
use specific prompts and follow the original test prompts.

## D.7 GSM8K

The GSM8K (Grade School Math 8,000) benchmark [8] serves as a rigorous evaluation framework
for testing the mathematical problem-solving prowess of LLMs. This benchmark comprises a dataset
of 8,500 diverse and high-quality math word problems at the grade school level, designed to challenge

**Prompt**

Question:
You are an expert Python programmer, and here is your task:
{ few_shot_instruction }
 Your code should pass these tests:
{ few_shot_test_case }
Your code should start with a [BEGIN] tag and end with a [DONE] tag.
Answer:
{ few_shot_response }
Question:
You are an expert Python programmer, and here is your task:
{ few_shot_instruction }
 Your code should pass these tests:
{ few_shot_test_case }
Your code should start with a [BEGIN] tag and end with a [DONE] tag.
Answer:
{ few_shot_response }
Question:
You are an expert Python programmer, and here is your task:
{ few_shot_instruction }
 Your code should pass these tests:
{ few_shot_test_case }
Your code should start with a [BEGIN] tag and end with a [DONE] tag.
Answer:
{ few_shot_response }
Question:
You are an expert Python programmer, and here is your task:
{ few_shot_instruction }
 Your code should pass these tests:
{ few_shot_test_case }
Your code should start with a [BEGIN] tag and end with a [DONE] tag.
Answer:

Figure A2: Three-shot prompt used to evaluate on MBPP and MBPP+.

**Code Review Data Generation**

You are asked to act as a professional code reviewer and your task is to professionally and accurately review the given content and assign a score. The involved programming languages include but are not limited to Python, C, C++, Java, JavaScript, HTML, Haskell, SQL, C#, and PHP.

**Please adhere to the following review requirements:**
- **Correct:** No syntax and logic errors. The implementations should follow the given function names.
- **Clarity:** Variables should have meaningful names. The arguments and return values of functions should have type annotations.

***NOTE (important):***
1. Please evaluate and score the code from the aspects of 'correct' and 'clarity', and adhere to the following output format:
[Correct]
<your score (ranges from 0 to 10)>. <your review>
[Clarity]
<your score (ranges from 0 to 10)>. <your review>
[Suggestions]
<your additional suggestions for further improvement>
[Refined Code]
<The refined code according to your suggestions (DO NOT modify the function declarations appearing in the code)>
2. If there is no code in the given content, please answer with [N/A].
3. DO NOT respond with content that is outside of the specified output format.

**Given Content:**
{{ code from multi-source data}}

Figure A3: Detailed prompt designed for generating code review data.

LLMs with tasks necessitating advanced, multi-step reasoning abilities. GSM8K's primary aim is to gauge how well these models can parse, understand, and solve math problems, thereby offering a comprehensive measure of their capacity for logical reasoning and mathematical computation. By incorporating such a specialized benchmark, researchers can better understand the extent to which LLMs can mimic human-like reasoning in solving complex mathematical scenarios. For evaluation on GSM8K, we do not use specific prompts and follow the original test prompts.

# E    AlchemistCoder Dataset Details

## E.1    *AlchemistPrompt*

We provide two samples of *AlchemistPrompt*s in Fig. A4 and Fig. A5.

## E.2    Code comprehension task data

For instruction evolution task data, we provide two samples in Fig. A6 and Fig. A7. For data filtering task data, we provide two samples in Fig. A8 and Fig. A9. For code review task data, we design prompt as illustrated in Fig. A3 to obtain high-quality code review task data and we provide two samples in Fig. A10 and Fig. A11.

Design a function that finds the greatest common divisor (GCD) of two given numbers: 10, 20.

**Response**

Sure, here is a Python function that uses the Euclidean algorithm to find the GCD of two numbers:

```python
def gcd(a, b):
    while b != 0:
        a, b = b, a % b
    return abs(a)

print(gcd(10, 20)) # Output: 10
```

This function keeps replacing `a` with `b` and `b` with `a % b` (the remainder of `a / b`) until `b` is `0`, at which point `a` is the greatest
common divisor. So, the GCD of 10 and 20 would be 10.

You can call the above function with any two numbers you want to find the GCD of.

*AlchemistPrompt*-customized **Instruction**

**Please provide a Python code example for the following task including the expected output after the code execution.**
Design a function that finds the greatest common divisor (GCD) of two given numbers: 10, 20.

Figure A4: Example #1 of *AlchemistPrompt*s.

**Instruction**

Compare two strings and output true if they are an anagram. The input strings can contain both lowercase and uppercase letters.

**Response**

```python
def is_anagram(string1, string2):
    string1 = string1.lower()
    string2 = string2.lower()
    string1 = string1.replace(" ", "")
    string2 = string2.replace(" ", "")

    list1 = list(string1)
    list2 = list(string2)
    list1.sort()
    list2.sort()

    sorted_string1 = ''.join(list1)
    sorted_string2 = ''.join(list2)

    # Compare the sorted strings
    if sorted_string1 == sorted_string2:
        return True
    else:
        return False
```

*AlchemistPrompt*-customized **Instruction**

**Write a Python function to check if two input strings are anagrams of each other, considering both lower and uppercase letters. An anagram is a word or phrase formed by rearranging the letters of a different word or phrase.**
Compare two strings and output true if they are an anagram. The input strings can contain both lowercase and uppercase letters.

Figure A5: Example #2 of *AlchemistPrompt*s.

**Instruction**

Please increase the difficulty of the given programming test question a bit.
You can increase the difficulty using, but not limited to, the following methods:
If the original problem can be solved with only a few logical steps, please add more reasoning steps.

**Given Prompt:**
Construct an SQL query to add a new column called "Gender" to the table "Employees" with values "Male" and "Female", but also make sure that the values for the "Gender" column are randomly assigned for each employee.

**Response**

Construct an SQL query to add a new column called "Gender" to the table "Employees" with values "Male" and "Female", but also make sure that the values for the "Gender" column are randomly assigned for each employee. Additionally, ensure that the ratio of male to female employees is approximately 1:1.

Figure A6: Example #1 of instruction evolution task data.

**Instruction**

Please increase the difficulty of the given programming test question a bit.
You can increase the difficulty using, but not limited to, the following methods:
If the original problem can be solved with only a few logical steps, please add more reasoning steps.

**Given Prompt:**
Iterate over a list and count the number of occurrences of each item. The list may contain duplicates and non-integer elements. Return a dictionary with the count of each item. The input list can have a maximum length of 1000.

**Response**

Iterate over a list and count the number of occurrences of each item. The list may contain duplicates and non-integer elements. However, now you need to also handle nested lists and dictionaries within the main list. You should recursively iterate through the nested data structures and count the occurrences of each item. Return a dictionary with the count of each item. Additionally, consider optimizing the code for performance, as the input list can have a maximum length of 10,000.

Figure A7: Example #2 of instruction evolution task data.

**Task:**
Before we proceed, let's briefly talk about Python list indexing. Python uses zero-based indexing, which means the first element of a list is at index 0. The list 'a' here is a mutable sequence of integers, and we are going to modify it using a for loop.
Consider a list `a = [1,3,2,2,3,4]`. What would be the output of the following code?

**Solution:**
The correct output of the given code would be: [3, 1, 3, 2, 2, 3]

Figure A8: Example #1 of data filtering task data.

Below is a code task and its corresponding solution. Please utilize your understanding of code to carefully review this solution.

**Task:**

Write a function that takes as input a tuple of numbers (t_1,...,t_{N+1}) and returns a tuple of length N where the i-th element of the tuple is equal to t_i * t_{i+1}. The function should also handle negative numbers and zeros, and return an error message if the input is not a tuple.

**Solution:**

Here is a function implementation in Python:

```python
def tuple_product(input_tuple):
    if not isinstance(input_tuple, tuple):
        return "Error: Input is not a tuple"
    return tuple(input_tuple[i] * input_tuple[i + 1] for i in
range(len(input_tuple) - 1))

print(tuple_product((1, 2, 3, 4, 5))
# Outputs: (2, 6, 12, 20)

print(tuple_product((-1, 2, -3, 4, -5)))
# Outputs: (-2, -6, -12, -20)
```

The above solution is considered to be of **\*low quality\***. Choose from the following four options what category this solution belongs to:
a. The solution is too short (less than 20 words) and does not contain code.
b. There are code compilation errors in the solution.
c. The clarity of the code in the solution is too poor.
d. The solution contains multiple sections of code and is not organized into modularized functions.

**Your choice (answer only the choice without the reason):**

b. There are code compilation errors in the solution.

Figure A9: Example #2 of data filtering task data.

## Instruction

*The prompt for generating data is omitted.*

**Given Content:**

You can use the `generate_json` function to get JSON string:

```python
import json
def process_value(value):
    if isinstance(value, str):
        return value.upper()
    elif isinstance(value, int):
        return value * 2
    elif isinstance(value, list):
        return [process_value(element) for element in value]
    elif isinstance(value, dict):
        return {key.upper(): process_value(val) for key, val in
value.items()}
    else:
        return value
def generate_json(dictionary):
    modified_dictionary = process_value(dictionary)
    json_string = json.dumps(modified_dictionary)
    return json_string
```

## Response

**[Correct]**

9. The code is correct. It correctly processes the values according to the specified rules and returns the expected JSON string.

**[Clarity]**

8. The code is clear and easy to understand. The function names are descriptive and the code follows a logical flow.

**[Suggestions]**

- It would be helpful to include type annotations for the function arguments and return values. This would make the code more clear and easier to understand.

**[Refined Code]**

```python
import json
from typing import Any, Dict, List, Union

def process_value(value: Union[str, int, List[Any], Dict[str, Any]]) ->
Union[str, int, List[Any], Dict[str, Any]]:
    if isinstance(value, str):
        return value.upper()
    elif isinstance(value, int):
        return value * 2
    elif isinstance(value, list):
        return [process_value(element) for element in value]
    elif isinstance(value, dict):
        return {key.upper(): process_value(val) for key, val in
value.items()}
    else:
        return value

def generate_json(dictionary: Dict[str, Any]) -> str:
    modified_dictionary = process_value(dictionary)
    json_string = json.dumps(modified_dictionary)
    return json_string
```

Figure A10: Example #1 of code review task data. For clarity, the prompt for generating data is omitted.

**Instruction**

*The prompt for generating data is omitted.*

**Given Content:**

The error in the code is an incorrect indentation when printing the indentation spaces. The number of spaces printed should be based on the line number, but the code mistakenly indents by a fixed number of spaces. This leads to misalignment and missing characters in some lines. Additionally, there are missing asterisks and periods in the inner loop, which causes the alternating pattern to be incorrect. By fixing the indentation and adding the missing characters, the tower of stars will be displayed correctly.

```python
def print_star_tower():
    for i in range(1, 11):
        # Error: Incorrect indentation
         print(' ' * i, end='') # Indentation should be based on line number
        for j in range(1, i*3):
            if j % 2 == 0:
                print('.', end='') # Error: Missing asterisk
            else:
                print('*', end='') # Error: Missing period
        print('')

print_star_tower()
```

**Response**

**[Correct]**

3. The code has syntax and there are logic errors that cause the tower of stars to be displayed incorrectly.

**[Clarity]**

8. The code uses meaningful variable names and is easy to understand.

**[Suggestions]**

- The code could benefit from adding type annotations to the function and its arguments.
- The logic errors could be fixed by adjusting the indentation and adding the missing characters in the inner loop.

**[Refined Code]**

```python
def print_star_tower() -> None:
    for i in range(1, 11):
        print(' ' * i, end='')
        for j in range(1, i*3):
            if j % 2 == 0:
                print('.', end='')
            else:
                print('*', end='')
        print()
```

In the refined code, the indentation has been fixed and the missing characters have been added. Additionally, type annotations have been added to the function and its arguments to improve clarity.

Figure A11: Example #2 of code review task data. For clarity, the prompt for generating data is omitted.

