# OpenReview forum: "AlchemistCoder: Harmonizing and Eliciting Code Capability by Hindsight Tuning on Multi-source Data"
_NeurIPS.cc/2024/Conference — NeurIPS 2024 poster_

### Official Review · Reviewer_FM86 · 2024-07-13

**Soundness:** 3
**Presentation:** 2
**Contribution:** 3
**Rating:** 6
**Confidence:** 3

**Summary:**

This paper presents an improved pipeline for training LLMs for code generation. Their method incorporates prompt modifications using hindsight tuning to modify prompts to better align with the associated code. In addition, they introduce improved data filtering and additional training tasks which they find give moderate additional performance boosts. Models trained using their pipeline show better generation abilities than existing LLMs and code LLMs on existing benchmarks and retain better ability on natural language tasks

**Strengths:**

The method is relatively simple (though somewhat costly) and appears to give performance boosts across tasks. Improving the performance of Code LLMs in ways beyond scaling up data is a valuable contribution to the community and this method has demonstrated utility. The authors compare to a variety of models and demonstrate the efficacy of their method on a variety of model sizes and families. Additionally, the authors give good ablation studies on what elements in their method contribute to this performance boost.

**Weaknesses:**

The main weaknesses of the paper come in lack of clarity. If these points can be clarified, I would be inclined to raise my score:

1. The writing in this paper was not especially clear to me, particularly when it came to motivating introducing this method. When contrasting with prior work, it would help to give examples of the type of low code quality and diversity that has been observed. More concerningly, because these details were not clearly explained, it was not clear to me how this dataset improved upon prior work.

2. The selection criteria are unclear for what samples that were chosen to be improved using AlchemistPrompts. If it is only a random 5% of the data that needs to be improved, it is unclear to me why performance benefits do not continue for higher percentages. On the other hand, if the data is selected more specially for this, it should be made clearer.

**Questions:**

1. Is the 5% of data modified by AlchemistPrompter chosen at random or targeted? If they were chosen randomly, was the experiment repeated? Finally, if they were chosen randomly, is there a chance that using a heuristic to choose them could improve performance further?

2. Why should we expect AlchemistCoder models to be better generalists in NL tasks when the diversity they have seen is mostly in the form of code data? Though there is more diversity in this dataset, it is still domain specific, which makes the catastrophic forgetting hypothesis seem unlikely to me. Do you perform any further experiments on this?

**Limitations:**

The authors explain adequately.

---

> ### Author Rebuttal · Authors · 2024-08-07
>
> **W1: Concerns of the presentation.**
> - Thank you for your valuable advice on improving our presentation! Our motivation stems from the following two considerations: 1) Existing Code LLM pre-train methods typically use multi-source data, while Code LLM fine-tuning methods focus on developing high-quality single-source datasets; 2) The current open-source Code LLM community already has various high-quality single-source fine-tuning data. To better utilize existing data resources and contribute to the Code LLM community, we are pioneering the integration of multi-source data for Code LLM fine-tuning.
> - Additionally, we have provided more examples in the global response PDF that demonstrate how our method harmonizes inherent conflicts in multi-source data. We will also refine our presentation in the latest version of the manuscript.
>
> **W2&Q1: Concerns about the selection of data customized by AlchemistPrompt.**
> - Sorry for the confusion. We calculate the Conditional Perplexity Discrepancy (CPD, refer to lines 248-252 of the manuscript) and **selectively chose data with higher CPD values for AlchemistPrompts harmonization**. Conditional perplexity discrepancy is an indicator of how data affects the complexity of model-generated responses under given conditions (i.e., instructions), and its calculation formula is $CPD = Perplexity(instruction + response) − Perplexity(response)$. The level of CPD reflects the impact of the conditional instruction on the complexity of the generated response. Specifically, a high CPD indicates that the perplexity of the generated response significantly increases under the presence of a conditional instruction, which usually reflects a poor alignment between the instruction and the response, the instruction may be unclear or not specific enough, or insufficient contextual information, thereby increasing the difficulty of model response generation. By analyzing high CPD values, we can identify cases where instructions and responses are poorly aligned and more effectively optimize data quality. As deeply analyzed in Figure 8 of the manuscript, AlchemistPrompts can effectively harmonize the discrepancy between instructions and responses.
>
> **Q2: Why should we expect AlchemistCoder models to be better generalists in NL tasks?**
> - Indeed, similar findings are also present in existing work in some other fields. Based on our insights, we attribute this phenomenon to the following three points:
>   - **Enhanced reasoning abilities**: Coding abilities themselves reflect reasoning capabilities. Additionally, multi-source integration significantly increases the diversity of fine-tuning data, which often includes detailed analyses, reasoning, and explanatory annotations, all of which contribute to improved model reasoning abilities.
>   - **Better instruction-following abilities**: AlchemistPrompts supplements instructions with details about programming languages, algorithm concepts, and code characteristics corresponding to the responses, representing an optimization of instruction/response alignment specific to coding capabilities. Thus, AlchemistPrompts can refine the alignment within instruction-response pairs and enhance the instruction-following abilities of fine-tuned models.
>   - **Improved context understanding**: As shown in Figure A4~A11 of the manuscript, both our AlchemistPrompts and code comprehension tasks can provide training for the context-understanding capabilities of models.
> - In summary, our fine-tuning data includes a substantial amount of natural language descriptions in addition to code snippets, leading to the improvements from fine-tuning that are not limited to coding abilities. The enhancements in the three abilities mentioned above also benefit tasks MMLU for multitasking language understanding, BBH for comprehensive reasoning, and GSM8K for mathematical reasoning.

---

> > ### Comment · Reviewer_FM86 · 2024-08-12
> >
> > Apologies for my late reply. Thank you for the clarification regarding the motivation. The example given in your response PDF of the possible mismatches was helpful, as was the point about how you chose the samples with AlchemistPrompt. Given this, and assuming that these changes are incorporated into the paper, I'm willing to increase my score.

---

> > > ### Author Response · Authors · 2024-08-13
> > >
> > > Thank you for increasing your score! Your insights have been incredibly helpful, and we are excited to incorporate the changes based on your suggestions into our paper.
> > >
> > > Thanks again for your support and valuable input!

---

### Official Review · Reviewer_yLAq · 2024-07-15

**Soundness:** 3
**Presentation:** 4
**Contribution:** 3
**Rating:** 6
**Confidence:** 3

**Summary:**

The paper presents a series of Code Large Language Models (LLMs) named AlchemistCoder, which are fine-tuned on multi-source data to enhance code generation and generalization capabilities. The authors address the limitations of previous Code LLMs that were typically fine-tuned on single-source data, which lacked diversity and quality, by introducing AlchemistPrompts, data-specific prompts generated through hindsight relabeling to harmonize different data sources and improve instruction-response pairs. Additionally, they incorporate the data construction process into fine-tuning as code comprehension tasks, including instruction evolution, data filtering, and code review. Extensive experiments demonstrate that AlchemistCoder outperforms models of the same size and rivals or surpasses larger models, showcasing its efficacy in refining instruction-following capabilities and advancing code intelligence.

**Strengths:**

1. The paper is clear and well-written.
2. The contribution is simple, clear, and easy to be adopted. (Use GPT4-generated prompts to harmonize the domain gap)
3. The evaluation justifies the effectiveness of the proposed method. (The ablation study explains the effectiveness of each step.)
4. The improvement is impressive.

**Weaknesses:**

I'm not fully convinced that the improvement is from "multi-source" data. AlchemistPrompts possibly introduced high-quality data from GPT-4. Table 4 also confirmed that the most gain is from that 5% AlchemistPrompt.
If this is the reason, the comparison with other models weaker than GPT4 is not completely fair.
It also means the proposed method is weak in relying on a strong model to supervise.

**Questions:**

For different portions of AlchemistPrompts other than 5%, does the general increasing trend of accuracy on multi-source data in Table 4 maintained?

**Limitations:**

See weaknesses.

---

> ### Author Rebuttal · Authors · 2024-08-07
>
> **W1: Concerns of AlchemistPrompts.**
> - Thanks for your insightful concerns! To fairly compare with other models, we provide a new version of Table 1 in the global response PDF, which includes details of training corpora sources for fine-tuned Code LLMs. Compared to methods that heavily rely on GPT-4 to generate entire new datasets, we strive to minimize our dependence on GPT-4 and only generate AlchemistPrompts for 5% of the data. **More importantly, existing Code LLM methods usually use strong models (e.g., GPT-4) to directly generate code for fine-tuning, whereas we do not. The AlchemistPrompts we generate are concise textual descriptions that do not include code (refer to Figures A4 and A5 in the manuscript), fundamentally differing from other methods in both their intended goals and practical effects.** To sum up, we do not rely directly on the code capabilities of strong models for supervision and our method achieves more effective optimization of fine-tuning data with greater diversity, higher quality, and lower cost for empowering AlchemistCoder to obtain promising and comprehensive code capabilities.
> - Additionally, we highlight the importance of combining AlchemistPrompts with multi-source data to achieve exceptional performance and we present more examples in Figure R1 of the global response PDF that demonstrate how our method harmonizes inherent conflicts in multi-source data. This is fundamentally different from fine-tuning on a single-source high-quality dataset. The synergy between AlchemistPrompts and multi-source data sets AlchemistCoder apart from other models and demonstrates a new path for enhancing code LLMs. Our contributions also lie in introducing new insights into improving prompts and designing instruction fine-tuning tasks to develop better open-source models: 1) The design philosophy and utility of AlchemistPrompts (please refer to our response to Reviewer q6UB W1); 2) The data construction process itself reflects higher-level capabilities and can guide model training.
> - Indeed, GPT-4 holds advantages in code and general capabilities and this urges us to continue advancing towards our goal: exploring techniques to optimize large models and bridging the gap between open-source and closed-source models. We will open source our fine-tuning data to contribute to the Code LLM community and explore the expansion of the proposed method to other LLM tasks.
>
> **Q1: For different portions of AlchemistPrompts other than 5%, does the general increasing trend of accuracy on multi-source data in Table 4 maintained?**
> - Yes, although 5% of AlchemistPrompts significantly contribute to performance improvement, other proportions (1–20%) of AlchemistPrompts also maintain a similar trend of enhancement. It is important to note that code data from different sources may vary greatly in language style and content, including question types, code style, presence of comments, test cases, etc. Therefore, mixing multi-source data has a double-edged sword effect: it provides necessary diversity but may also introduce significant domain gaps. To effectively bridge this gap while maintaining diversity, adding concise corpus generated by the same Alchemist model (i.e., AlchemistPrompts with similar language styles) to a small amount of data can effectively address this issue. Additionally, AlchemistPrompts are beneficial for refining the alignment within instruction-response pairs to enhance the instruction-following abilities of fine-tuned models.

---

> > ### Author Response · Authors · 2024-08-13
> > **Please let us know if your concerns have been addressed**
> >
> > Dear Reviewer yLAq,
> >
> > We wish to express our gratitude for your thorough review and positive feedback. Likewise, **we are warmly concerned whether our rebuttal addresses your concerns**. Your feedback is invaluable to us, and we are fully committed to thoughtfully incorporating your insights to enhance our paper.
> >
> > Once again, thank you for your ongoing support during this review process!
> >
> > Sincerely,
> >
> > Authors of Paper #10234

---

### Official Review · Reviewer_fYHy · 2024-07-21

**Soundness:** 4
**Presentation:** 3
**Contribution:** 2
**Rating:** 6
**Confidence:** 3

**Summary:**

This work improves upon past work developing code LLMs with a focus on intervention on the data used to instruction tune the models. Specifically, the key insight in this work is that past works have usually relied on single source data for fine-tuning, but this can come at a drawback of quality and diversity. To reduce these issues, the authors use multi-source data. This in turn leads to a challenge where the same question can elicit multiple responses, calling for the need for AlchemistPrompts that "harmonize" the sourced. A second source of gains for the AlchemistCoder models series comes from construction of code comprehension tasks. Various experiments are performed to demonstrate the method is very effective in achieving performance as good as models from a larger model size.

**Strengths:**

1. Clear problem identification: I liked that the authors identified a problem in existing literature training code LLMs pertaining to single source code data. They performed subsequent steps to mitigate this gap with single-source data, then harmonizing it.
2. Strong results: The alchemistcoder model series punches much above models of the same parameter size, as evident from results in Table 1 and Figure 1.
3. Wide range of evaluations: The authors test their model capabilities on both code generation tasks, and standard benchmarks, and the performance gains stay across the board.
4. Analytical experiments on data composition: I liked the analysis in Figures 5, 6 and 8 giving more perspective to the reader on the differences between AlchemistCoder data and data from past work.

**Weaknesses:**

1. Missing ablations: Many of the comparisons are nor iso-compute (different models trained for different time). I believe an important aspect of this work is to clearly show model performance w.r.t. various ablations performed: 1. Multi-source 2. +Harmonization. 3. +Code comprehension.
2. Python-based evals: Even though multi-source data is used, all the evaluations are limited to python based evaluation. I would have liked to see more analysis on how the multi-source data impacts model performance in other languages.

**Questions:**

1. Can you point me to this study: "Our ablation study indicates that the optimal performance can be achieved by incorporating AlchemistPrompts into only 5% of all the samples, striking a balance between the diversity and domain gap resulting from the fusion of multi-source data."
2. Is code comprehension data augmentation a novel contribution of this work?

**Limitations:**

The generation of AlchemistPrompts relies heavily on GPT-4

---

> ### Author Rebuttal · Authors · 2024-08-07
>
> **W1: Missing ablations.**
> - Thanks for your meticulous suggestions! We provide a new version of Table 1 in the global response PDF, which includes details of the training corpus used for fine-tuned Code LLMs.
> - Here, we reorganize the following table based on Table 4 to more clearly demonstrate the impact of multi-source data, harmonization, and code comprehension:
>   |**Method**|**HumanEval (Pass@1)**|**MBPP (Pass@1)**|
>   |-|:-:|:-:|
>   |Baseline (CodeLlama-Python-7B)|37.8|57.6|
>   |+ Multi-source data (w/o data decontamination)|54.6|57.9|
>   |+ Multi-source data (w data decontamination)|59.8|58.2|
>   |+ Harmonizations (AlchemistPrompts)|72.0|63.4|
>   |+ Code comprehension (Instruction Evolution)|71.3|65.8|
>   |+ Code comprehension (Data Filtering)|73.8|67.7|
>   |+ Code comprehension (Code Review)|**74.4**|**68.5**|
> - Due to inherent conflicts, the improvement brought by directly mixing multi-source data is limited (refer to the second/third row of the table above and DirectMix-L-7B in Figure 1). We believe that the key efficacy of our method lies in optimizing the data to possess the following characteristics conducive to effective model training:
>   - **Balance between diversity and source gap**: Multi-source mixing may bring necessary diversity but harmful domain gaps. Adding concise corpus generated from the same Alchemist model (i.e., AlchemistPrompts with similar language styles) can effectively bridge this gap.
>   - **Better alignment within instruction-response pairs**: AlchemistPrompts supplements instructions with details about programming languages, algorithm concepts, and code characteristics corresponding to the responses, representing an optimization of instruction/response alignment specific to coding capabilities. As shown on the right side of Figure 8, the instructions customized by AlchemistPrompts are closer to the responses in the feature space. Meanwhile, as shown on the left side of Figure 8, Conditional Perplexity Discrepancy (CPD) quantifies the difficulty change in generating responses before and after adding specific inputs (e.g., instructions) to the model (the smaller the value, the easier it becomes). The generally smaller CPD values after adding AlchemistPrompts reflect the facilitating effect of AlchemistPrompts on improving model performance.
>   - **Reflecting code comprehension**: Instruction evolution task data, which communicates the relationship between data before and after evolution, addressing a gap in existing work using instruction evolution data. Data filtering and code evaluation task data, provide the model with a deep understanding of both high-quality and low-quality code.
> - In addition, we have explored which data features are harmful to model training (refer to the fourth row of the table above and lines 162-171 of the manuscript):
>   - Responses that are overly concise and devoid of code. These answers usually provide straightforward replies to the instructions, overlooking both the code solution and explanatory annotations. Moreover, these instances often contain overly simplistic questions in the instructions.
>   - Code solutions that are either non-compilable or do not pass test cases (pertaining to particular samples).
>
> **W2: Python-based evals.**
> - Indeed, similar to existing work, we focus on Python due to its popularity and the wealth of benchmark resources available. We have provided results on multilingual code generation (refer to Table 2 in the manuscript, including C++, Go, Java, and JavaScript) and results on general benchmarks (refer to Table 5 in the manuscript, including MMLU for multitask language understanding, BBH for comprehensive reasoning, and GSM8K for mathematical ability). These extensive experimental results demonstrate that our AlchemistCoder series models deliver comprehensive code capabilities and reasoning abilities.
> - Additionally, each single source data itself contains multiple programming languages, not just one language per source, making it challenging to separately ablate the corpus for each language. Under the efficacy of multi-source integration, our fine-tuning data includes nearly 60% of various programming languages (refer to Figure 5 in the manuscript, including C, Java, HTML, SQL, etc.), aiming to contribute more universally powerful open-source models to the Code LLM community.
>
> **Q1: Clarification of the ablation study.**
> - Sorry if that statement was confusing. Code data from different sources may vary significantly in language style and content, including question types, code style, presence of comments, test cases, etc. Therefore, multi-source data mixing is a double-edged sword: it provides necessary diversity but can also bring large domain gaps. Adding concise corpus generated from the same Alchemist model (i.e., AlchemistPrompts with similar language styles) to a small amout of data can effectively bridge this gap while maintaining diversity. Besides, the inclusion of 5% AlchemistPrompts also represents a balance between cost and performance.
>
> **Q2: Is code comprehension data augmentation a novel contribution?**
> - Yes, we pioneer novel code comprehension tasks for fine-tuning Code LLMs. Meanwhile, we also aim to provide new insights into the construction of fine-tuning data: the data construction process itself reflects higher-level capabilities and can guide model training.
>
> **L1: The Reliance on GPT-4.**
> - Our method does indeed rely on Alchemist models (i.e., GPT-4), and we have discussed this in section A of limitations. Compared to methods that heavily rely on GPT-4 to generate entire new datasets, we strive to minimize our dependence on GPT-4 and only generate concise AlchemistPrompts for 5% of the data.
> - More importantly, this urges us to continue advancing toward our goal: exploring techniques to optimize large models and bridging the gap between open-source and closed-source models. And we will open source our fine-tuning data to contribute to the Code LLM community.

---

> > ### Author Response · Authors · 2024-08-13
> > **Please let us know if your concerns have been addressed**
> >
> > Dear Reviewer fYHy,
> >
> > We wish to express our gratitude for your extensive review and positive feedback. Likewise, **we are warmly concerned whether our rebuttal addresses your concerns**. Your feedback is invaluable to us, and we are fully committed to thoughtfully incorporating your insights to enhance our paper.
> >
> > Once again, thank you for your ongoing support during this review process!
> >
> > Sincerely,
> >
> > Authors of Paper #10234

---

### Official Review · Reviewer_q6UB · 2024-07-24

**Soundness:** 3
**Presentation:** 3
**Contribution:** 2
**Rating:** 6
**Confidence:** 3

**Summary:**

This paper introduces AlchemistCoder, a series of code language models fine-tuned on multi-source data. The authors propose using "AlchemistPrompts" to harmonize inherent conflicts in multi-source code corpora and incorporate code comprehension tasks into the training process. The resulting models show improved performance on various code generation benchmarks compared to baseline models of similar size.

**Strengths:**

1. The paper addresses an important challenge in fine-tuning code language models by utilizing multi-source data, which could potentially lead to more robust and versatile models.
2. Empirical results demonstrate significant improvements over baseline models on several code generation benchmarks, including HumanEval, MBPP, and DS-1000. The authors provide detailed ablation studies and analyses to validate the effectiveness of their proposed methods, including the impact of AlchemistPrompts and code comprehension tasks.

**Weaknesses:**

1. The core concept of AlchemistPrompts lacks substantial novelty. It appears to be a variation of existing instruction evolution techniques, such as the Evol-Instruct.
2. While the performance improvements are notable, the overall approach of using multi-source data and harmonizing it is not fundamentally new in the field of language model fine-tuning.
3. The paper compares AlchemistCoder models (6.7B/7B parameters) with larger models, including 15B, 33B, and 70B models. While the results show that AlchemistCoder outperforms or rivals these larger models on several benchmarks (as seen in Table 1 and Figure 1), the paper does not provide a comprehensive analysis of the factors contributing to this performance gain. It would be valuable to have a more in-depth discussion on how much of this improvement can be attributed to the proposed method versus other potential factors such as the efficiency of smaller models or the quality of the base model used. This additional analysis would strengthen the paper's claims about the effectiveness of the AlchemistCoder approach relative to simply scaling up model size.

**Questions:**

1. Can you provide more insights into the relative contributions of data diversity and the harmonization process to the overall performance improvement? Are there scenarios where one factor might be more important than the other?
2. How does the efficiency and effectiveness of the AlchemistPrompt approach scale with increasing model size and dataset complexity? Are there any computational or performance bottlenecks that might limit its applicability to very large language models or extremely diverse multi-source datasets?

**Limitations:**

The authors have addressed some limitations of their work, such as the potential for hallucination in response to obscure queries. However, there are several other important limitations that could be more thoroughly discussed:
1. Scalability: The paper focuses on models with 6.7B/7B parameters. There's limited discussion on how the AlchemistPrompt approach and code comprehension tasks would scale to much larger models or more diverse datasets. The computational costs and potential performance bottlenecks for scaling up are not addressed.
2. Data Bias: The process of selecting and harmonizing multi-source data could potentially introduce or amplify biases present in the original datasets. The paper doesn't thoroughly address how these biases are identified, mitigated, or might impact the model's outputs.

---

> ### Author Rebuttal · Authors · 2024-08-07
>
> **W1&W2: The novelty of the proposed method.**
> - Actually, our AlchemistPrompts are entirely different from existing instruction evolution techniques in several aspects:
>   - **Different designed goals**: Instruction evolution techniques are designed to "expand into a richer and more complex set of instructions," whereas the goal of AlchemistPrompts is to harmonize multi-source data and instruction-response pairs. In short, the former aims at "**expansion**", while the latter focuses on "**harmonization**".
>   - **Different operational methods**: Instruction evolution techniques start from an initial set of instructions **only** and typically involve multiple rounds of **iterative** generation for **all** data. In contrast, AlchemistPrompts use only a **small** amount of data, require **both** instructions and responses, and only need to generate **once**. Additionally, instruction evolution techniques require generating **new** responses, whereas our AlchemistPrompts do **not**.
>   - **Different practical effects**: In terms of **diversity**, instruction evolution techniques significantly expand the data, while AlchemistPrompts suppress excessive diversity in multi-source data by incorporating corpora with similar language styles. Regarding **controllability**, instruction evolution techniques generate data with reference to a broad evolutionary direction, resulting in relatively high randomness, while AlchemistPrompts focus on fine-grained alignment within individual instruction-response pairs, thereby providing stronger controllability. For **data alteration**, AlchemistPrompts only insert concise corpus into the instructions, whereas instruction evolution techniques make substantial changes to both instructions and responses.
> - Additionally, our AlchemistPrompts explore a novel application of hindsight relabeling in the domain of Code LLM fine-tuning, which differs fundamentally from previous methods. Specifically:
>   |**Method**|**Designed Purpose**|**Relabeled Object**|**Generation Mode**|**Relabeling Period**|**Experience Source**|
>   |-|:-:|:-:|:-:|:-:|:-:|
>   |Previous Methods [a,b,c,d]|Alignment for Preferences|Conditional Goal|Handcrafting/Scripting|Postprocessing| Human|
>   |AlchemistPrompts (ours)|Harmonization for Multi-source Data|Data Instruction|LLM Generation|Preprocessing|LLM|
> - More importantly, our core contributions lie in pioneering the application of multi-source data **in the field of Code LLM fine-tuning** and we are **the first to unveil inherent conflicts in multi-source code corpora**. To achieve this, we propose logically progressive components of our method: AlchemistPrompts and code comprehension tasks. These key ideas introduce new insights into improving prompts and designing instruction fine-tuning tasks, enabling our fine-tuning data to be **more diverse, higher quality, and lower cost** for empowering AlchemistCoder with promising and comprehensive code capabilities.
>
> #### **Reference**
> [a] Andrychowicz M, et al. Hindsight experience replay. NeurIPS 2017.
>
> [b] Li A, et al. Generalized hindsight for reinforcement learning. NeurIPS 2020.
>
> [c] Packer C, et al. Hindsight task relabelling: Experience replay for sparse reward meta-rl. NeurIPS 2021.
>
> [d] Korbak T, et al. Pretraining language models with human preferences. ICML 2023.
>
> **W3: Additional analysis on the effectiveness of the proposed method.**
> - Thanks for your considerate suggestions! For additional analysis on the effectiveness of our method, please refer to our response to Reviewer fYHy W1.
>
> **Q1: More insights into the relative contributions of data diversity and the harmonization process.**
> - Code data from different sources may vary significantly in language style and content, including question types, code style, presence of comments, test cases, etc. Therefore, multi-source data mixing is a double-edged sword: it provides necessary diversity but can also bring large domain gaps. Adding concise corpus generated from the same Alchemist model (i.e., AlchemistPrompts with similar language styles) to a small amount of data can effectively bridge this gap while maintaining diversity.
> - In terms of balancing diversity and harmonization, for single-source data, the relatively lacking diversity is more important; for multi-source data, diversity is naturally introduced during mixing, so harmonizing conflicts becomes more crucial.
>
> **Q2&L1: Scalability & Data Complexity**
> - Thanks for your insightful advice! For scalability, due to limited computational resources, we focus on smaller models that are more popular and practical in the Code LLM field. For larger models, we will discuss this in our limitations and will open source our fine-tuning data to contribute to the Code LLM community, welcoming developers to apply it to larger models.
> - In terms of data complexity, we assume that an increase in data complexity implies the inclusion of more diverse data sources. We deconstruct this into a higher degree of (possibly excessive) diversity and more varied quality. In such a scenario, the harmonization effect introduced by our method becomes increasingly critical, which may imply a greater need for a higher proportion of AlchemistPrompts to achieve optimal performance. As for extreme cases, this essentially means fine-tuning towards generalist models rather than specialized ones (e.g., Code LLMs), where the bottleneck tends to be the balance of various capabilities, i.e., like a seesaw.
>
> **L2: Data Bias**
> - Thank you for emphasizing the aspect of data bias. Our AlchemistPrompts enhance the instruction-following abilities of models, which can potentially mitigate biases. For example, if the model has a bias towards responding with Python code when the programming language is not specified, the inclusion of programming language declarations in AlchemistPrompts helps to alleviate this bias. We have not delved into this yet and will discuss it in our limitations.

---

> > ### Comment · Reviewer_q6UB · 2024-08-12
> > **Thanks for the response**
> >
> > Thanks for the effort. I am willing to raise my score to 5 because the rebuttal effectively clarifies the unique aspects of AlchemistPrompts, distinguishing them from existing instruction evolution techniques. The authors' explanation of the different goals, methods, and practical effects is convincing. While I still have some minor questions about data complexity impacts, the overall approach seems promising for improving Code LLM fine-tuning with multi-source data.

---

> > > ### Author Response · Authors · 2024-08-13
> > >
> > > Thank you for raising the score and recognizing our response for clarifying your concerns!
> > >
> > > Your question about data complexity impacts is insightful and we believe a discussion around this would improve our paper. Data complexity is a fundamental issue in the domain of large models, affecting not only model training and performance but also data processing strategies, model generalization, and resource utilization. Specifically:
> > > - **Data Complexity and Multi-Source Integration**: Our research demonstrates that integrating data from multiple sources significantly increases data complexity and diversity, as evidenced by the broader distributions of code and description lengths shown in Figure 6 of the manuscript. While this integration facilitates the model's ability to learn richer feature representations, it also heightens the demands on the model to manage inputs of varying styles, formats, and quality. AlchemistCoder addresses this challenge by introducing AlchemistPrompts, which help conduct harmonizations across various data sources and within instruction-response pairs. To offer a more in-depth analysis of the AlchemistPrompt efficacy as data complexity scales, we present detailed experimental results from the multi-source integration and harmonization process:
> > >
> > >   | **Method**                                               | **HumanEval (Pass@1)** | **MBPP (Pass@1)** |
> > >   |----------------------------------------------------------|:----------------------:|:-----------------:|
> > >   | Baseline   (Llama2-7B)                                   |          14.0          |        26.1       |
> > >   | One-source data   fine-tuning (w data decontamination)   |          18.3          |        29.0       |
> > >   | + Harmonizations   (AlchemistPrompts)                    |       22.6 (4.3 $\uparrow$)       |     30.2 (1.2 $\uparrow$)    |
> > >   | Two-source data   fine-tuning (w data decontamination)   |          35.4          |        30.6       |
> > >   | + Harmonizations   (AlchemistPrompts)                    |       39.0 (3.6 $\uparrow$)       |     32.8 (2.2 $\uparrow$)    |
> > >   | Three-source data   fine-tuning (w data decontamination) |          37.8          |        35.4       |
> > >   | + Harmonizations   (AlchemistPrompts)                    |       43.9 (6.1 $\uparrow$)       |     40.8 (5.4 $\uparrow$)    |
> > >   | Four-source data   fine-tuning (w data decontamination)  |          40.2          |        42.2       |
> > >   | + Harmonizations   (AlchemistPrompts)                    |       55.1 (**14.9** $\uparrow$)      |     49.4 (**7.2** $\uparrow$)    |
> > >   | + Code   comprehensions (i.e., AlchemistCoder-L-7B)      |     **56.7** (1.6 $\uparrow$)     |   **54.5** (5.1 $\uparrow$)  |
> > >
> > > - **Data Cleaning and Decontamination**: In the face of complex data, preprocessing steps become crucial. Data cleaning and decontamination can remove noise and irrelevant information, helping the model focus on learning meaningful patterns. More specifically, we have explored which data features are harmful to model training (refer to the fourth row of the table in response to Reviewer fYHy W1 and lines 162-171 of the manuscript):
> > >   - Responses that are overly concise and devoid of code. These answers typically provide straightforward replies to instructions, neglecting both the code solution and explanatory annotations. Additionally, these instances often contain overly simplistic questions in the instructions.
> > >   - Code solutions that are either non-compilable or do not pass test cases (pertaining to specific samples).
> > > - **Model Generalization**: Data complexity also affects the generalization abilities of models. The models need to be adequately trained on complex data to perform well on unseen data. AlchemistCoder series models maintain enhanced code generation and generalization capabilities through multi-source data fine-tuning, thereby improving performance across a range of tasks.
> > > - **Cost and Efficiency Trade-Off**: Processing complex datasets may require more computational resources and time, creating a trade-off between cost and model performance. We strive to achieve significant performance improvements with only generating concise AlchemistPrompts for 5% of the data, thereby finding a balance between cost and efficiency.

---

> ### Author Response · Authors · 2024-08-14
> **Please let us know if your concerns have been addressed**
>
> Dear Reviewer q6UB,
>
> We wish to express our gratitude for your extensive review and supportive feedback. Your feedback of minor questions about data complexity impacts is invaluable to us, and **we are fully committed to thoughtfully incorporating your insights to enhance our paper**. As the discussion phase is nearing its end, we are warmly concerned whether our rebuttal addresses your concerns.
>
> **It would be appreciated if you could raise your score on our paper if we address your concerns**. We thank you again for your effort in reviewing our paper.
>
> Best regards,
>
> Authors of Paper #10234

---

> > ### Comment · Reviewer_q6UB · 2024-08-14
> > **Thanks for the detailed response**
> >
> > Sorry for the late reply. I am willing to raise my score to 6. The comprehensive analysis of data complexity impacts and multi-source integration addresses my concerns.

---

> > > ### Author Response · Authors · 2024-08-14
> > > **Thank you!**
> > >
> > > Thank you for raising the rating! Your feedback is invaluable, and we are delighted to make revisions to our paper based on your insights.

---

### Author Rebuttal · Authors · 2024-08-07

Dear all,


We appreciate the reviewers for valuable feedback remarking our work has "simple" method (**Reviewer yLAq & FM86**) and "strong" efficacy with "impressive improvements" (**Reviewer q6UB & fYHy & yLAq & FM86**), provides "wide range of evaluations, detailed ablation studies and analyses" (**Reviewer q6UB & fYHy & yLAq & FM86**), "identifies and addresses a important challenge of utilizing multi-source data for Code LLM fine-tuning" (**Reviewer q6UB & fYHy**), and is "clear and well-written" (**Reviewer fYHy & yLAq**). We have responded to all the concerns point by point and additional details mentioned in the response will be **synced** to the revised version of our manuscript.

The core innovation of our AlchemistCoder lies in proposing an effective framework for integrating multi-source data for Code LLM fine-tuning to overcome the limitations in quality and diversity inherent within a single-source dataset. **This is a non-trivial paradigm in the field of Code LLM fine-tuning and we are the first to unveil inherent conflicts in multi-source code corpora**. To resolve this challenge, we innovatively design data-specific AlchemistPrompts, inspired by hindsight relabeling. Additionally, we make the first effort to integrate the data construction process as code comprehension tasks into the training process. These key concepts facilitate the **enhancement of the diversity, quality, and cost-effectiveness** of our fine-tuning data, thereby enabling the development of the AlchemistCoder series models with significantly enhanced and comprehensive coding capabilities. Our contributions can be summarized as:
- **AlchemistPrompts**: Designed as data-specific prompts for harmonizing inherent conflicts in multi-source data and mitigating instruction/response misalignment at a fine-grained level.
- **Code Comprehension Tasks**: Sourced from data construction process, consisting of instruction evolution, data filtering, and code review.
- **Harmonized Multi-source Data**: Instruction tuned on 200M tokens, including 6 types of high-quality data.
- **Superior Model Performance**: Surpassing all the open-source models of the same size (6.7/7B), and rivaling or even beating larger models (15B/33B/70B/ChatGPT) on 6 code benchmarks.
- **Advanced Generic Capabilities**: Demonstrated by the significant improvements on MMLU, BBH, and GSM8K.

Once again, we really appreciate the supportive feedback and strongly believe that these reviews have strengthened the work.


Sincerely,

Authors of Paper #10234

---

### Author Response · Authors · 2024-08-12
**Gentle Reminder**

Dear ACs and Reviewers,

We wish to express our gratitude for your endeavors to participate in the review process. Our utmost efforts have been dedicated to resolving the concerns articulated by all reviewers. We sincerely hope that you have had a chance to review our response to your comments. We kindly request that you get back to us as soon as possible since the discussion period is ending soon. We appreciate your time and effort in reviewing our work, and we are happy to provide any further information if required.

Thank you for your ongoing support during this review process!

Sincerely,

Authors of Paper #10234

---

### Decision · Program_Chairs · 2024-09-25

**Decision:**

Accept (poster)

**Comment:**

The paper presents a series of Code LLMs called AlchemistCoder with an emphasis on multi-source data. In order to accommodate data sources with different styles, the authors design data-specific prompts with highsight relabeling. The empirical performance is strong despite that the authors focus on the models at around 7B parameters. The reviewers had questions about the 5% generated data but they were addressed by the authors.